# Single cell analysis reveals multiple requirements for zinc in the mammalian cell cycle

**Maria N Lo**[1,2], **Leah J Damon**[1,2], **Jian Wei Tay**[1,2], **Shang Jia**[3], **Amy E Palmer**[1,2]*

[1]Department of Biochemistry, University of Colorado, Boulder, Boulder, United States; [2]BioFrontiers Institute, University of Colorado, Boulder, Boulder, United States; [3]Department of Chemistry, University of California, Berkeley, Berkeley, United States

**Abstract** Zinc is widely recognized as essential for growth and proliferation, yet the mechanisms of how zinc deficiency arrests these processes remain enigmatic. Here we induce subtle zinc perturbations and track asynchronously cycling cells throughout division using fluorescent reporters, high throughput microscopy, and quantitative analysis. Zinc deficiency induces quiescence and resupply stimulates synchronized cell-cycle reentry. Monitoring cells before and after zinc deprivation we found the position of cells within the cell cycle determined whether they either went quiescent or entered another cell cycle but stalled in S-phase. Stalled cells exhibited prolonged S-phase, were defective in DNA synthesis and had increased DNA damage levels, suggesting a role for zinc in maintaining genome integrity. Finally, we demonstrate zinc deficiency-induced quiescence occurs independently of DNA-damage response pathways, and is distinct from mitogen removal and spontaneous quiescence. This suggests a novel pathway to quiescence and reveals essential micronutrients play a role in cell cycle regulation.

*For correspondence:
amy.palmer@colorado.edu

**Competing interests:** The authors declare that no competing interests exist.

## Introduction

Zinc ($Zn^{2+}$) is the second most abundant transition metal in biology and is widely recognized as an essential micronutrient to all living organisms (*Kaur et al., 2014*). $Zn^{2+}$ was first reported to be essential for growth of *Aspergillus niger* in 1869 and subsequently demonstrated for plants, animals, and humans (*Prasad, 1993*) with the first cases of human $Zn^{2+}$ deficiency and the associated growth and developmental disorders described in 1961 (*Prasad et al., 1961*). $Zn^{2+}$ deficiency has since been recognized as a global health problem, and the World Health Organization (WHO) estimates a staggering one third of the world's population does not consume adequate $Zn^{2+}$ and is therefore at risk for associated side effects and comorbidities (https://www.who.int/whr/2002/chapter4/en/index3.html) (*Roohani et al., 2013*). While the clinical manifestations of $Zn^{2+}$ deficiency are diverse and can be organism specific, one defining feature is universal: $Zn^{2+}$-deficient cells fail to divide and proliferate normally, leading to organismal growth impairment (*Vallee and Falchuk, 1993*). Despite recognition of the fundamental role of $Zn^{2+}$ for proliferation, the mechanisms of how $Zn^{2+}$ deficiency leads to cell-cycle arrest at the cellular and molecular level remain poorly defined.

Eukaryotic cell proliferation is governed by the cell-division cycle, a series of highly choreographed steps that involve gap (G1), DNA replication (S-phase), gap (G2), and mitosis (M) phases. Regulated transitions between proliferative and quiescent (i.e. reversible non-proliferative) states are essential for maintaining genome integrity and tissue homeostasis, ensuring proper development, and preventing tumorigenesis. Given the essentiality of $Zn^{2+}$ for growth and proliferation, a fundamental question is whether $Zn^{2+}$ serves as a nutrient, like amino acids, whether it affects the rate of cell cycle progression, or whether it is required at a specific phase of the cell cycle. Pioneering work

**eLife digest** For an animal to grow, its cells have to divide. Cell division can only take place if the cell meets certain conditions: for example, the cell's DNA must not be damaged. To ensure that cells only divide when these conditions are met, the cell goes through a series of stages or phases with checkpoints known as the cell cycle. The ability to control whether cells divide is essential for an animal to correctly form organs and tissues, or to heal wounds.

Zinc is a metal that animals get in their diet, and when zinc levels are low, animals usually grow more slowly because their cells stop dividing. Around 2 billion people worldwide do not get enough zinc in their diet. Amongst other processes, zinc is necessary for DNA repair, which could explain why low levels of zinc stop the cell from dividing. However, the evidence for why zinc is required for cell division is contradictory. Although it seems clear that zinc is necessary for cells to progress through the cell cycle, it was unknown whether it is needed at a specific stage or whether it influences a cell's decision to divide.

Lo et al. have used microscopy to examine the effects that different levels of zinc had on individual cells grown in the lab. The results suggest that cells can monitor the levels of zinc in their environment, and respond to low levels by shutting down growth and cell division. This happens independently of the stage of the cell cycle a cell finds itself in, explaining the discrepancies between older studies. Additionally, the results show that although zinc is required for DNA repair, this process is not what triggers cells to stop dividing in the absence of zinc. In fact, low levels of zinc seem to be stopping cell division through a previously unknown mechanism.

Lo et al.'s findings illustrate the high sensitivity of cells to changes in zinc availability and highlight the importance of zinc in our diet. Further study of how cells determine zinc levels and how those levels affect the cell cycle may help explain zinc's role in health and disease.

by Chesters et al sought to define precisely when $Zn^{2+}$ is required in the mammalian cell cycle. By chelating $Zn^{2+}$ at different timepoints after release from serum starvation-induced quiescence, they found that $Zn^{2+}$ was important for thymidine incorporation and thus DNA synthesis, leading to the conclusion that $Zn^{2+}$ was required for the G1 to S transition (*Chesters et al., 1989*). Subsequent studies confirmed that treatment of mammalian cells with high concentrations of metal chelators (DTPA and EDTA) seemed to compromise DNA synthesis (*Chesters et al., 1990*; *Chesters and Boyne, 1991*; *Watanabe et al., 1993*; *Prasad et al., 1996*). However, later studies by Chesters et al suggested that after cells passed the restriction point in mid-G1 there was no further $Zn^{2+}$ requirement for DNA synthesis in S phase, but rather $Zn^{2+}$ was needed to transition from G2/M back into G1 (*Chesters and Petrie, 1999*). The restriction point is classically defined as the point at which cells commit to completing the cell cycle, regardless of the presence of external growth factors such as mitogens and/or serum (*Pardee, 1974*). Thus, while these early studies suggested that $Zn^{2+}$ was important for progression of the mammalian cell cycle, the precise role of $Zn^{2+}$ and whether it is required at a specific stage have remained enigmatic.

There are three limitations of these early studies on the role of $Zn^{2+}$ in cell proliferation. First, because the analyses were carried out on populations of cells, the cells were synchronized by artificial means (serum starvation or hydroxyurea treatment) and the cell cycle phase was inferred based on release from the cell cycle block. Recently, it has become clear that synchronization can induce stress response pathways that are specific to the type of arrest (*Ly et al., 2015*; *Matson and Cook, 2017*; *Min and Spencer, 2019*). Further, cells induced into quiescence by different mechanisms (serum starvation, loss of adhesion, contact inhibition) exhibit overlapping but distinct transcriptional profiles, suggesting that different synchronization approaches impact cell cycle analysis upon emergence from quiescence (*Coller et al., 2006*). Second, population level analyses such as immunoblotting and qPCR mask cellular heterogeneity and subpopulations of cells with different cell fates and cell cycle dynamics (*Matson and Cook, 2017*; *Spencer et al., 2013*). Recent application of imaging and measurement tools for single cell analysis has uncovered distinct subpopulations of cells with different cell cycle dynamics (*Spencer et al., 2013*), and revealed key orders of molecular events in the decision between proliferation and quiescence (*Spencer et al., 2013*; *Cappell et al., 2016*; *Heldt et al., 2018*; *Moser et al., 2018*). Third, many of the previous investigations into the role of

$Zn^{2+}$ in the mammalian cell cycle have relied on high concentrations of chelators (DTPA, EDTA or TPEN) to induce $Zn^{2+}$ deficiency. However, these studies did not explicitly define how these perturbations changed the intracellular labile $Zn^{2+}$ pool, nor did they characterize how the chelators affected cell viability. Indeed, the concentrations of chelators used have been shown to induce apoptosis in a number of different cell types (*Sunderman, 1995*; *Johnson et al., 2000*; *Hyun et al., 2001*; *Kolenko et al., 2001*; *Canzoniero et al., 2003*; *Corniola et al., 2008*; *Lee et al., 2008*; *Makhov et al., 2008*; *Carraway and Dobner, 2012*; *Mendivil-Perez et al., 2012*; *Zhu et al., 2017*).

In this study, we revisit the fundamental and unresolved question of how $Zn^{2+}$ deficiency blocks cell proliferation using a combination of fluorescent reporters, high throughput microscopy, and quantitative image analysis. It is often overlooked that in addition to serving as a reservoir for mitogens, serum is also the major source of essential micronutrients including Zn, Fe, Cu, and Mn, and thus complete removal of serum also eliminates exogenous supply of these and other essential nutrients. By controlling $Zn^{2+}$ in the medium, while maintaining mitogens at levels that normally sustain proliferation, we induced subtle perturbations of labile $Zn^{2+}$ in the cytosol from 1 pM to 210 pM and tracked asynchronously cycling cells over multiple rounds of cell division. We found that $Zn^{2+}$ deficiency induces cellular quiescence, but not death, and $Zn^{2+}$ resupply stimulates synchronized cell cycle reentry. By following the entry of single cells into quiescence over time after $Zn^{2+}$ deprivation, we found that depending on where cells were in the cell cycle, they either entered quiescence immediately after mitosis, or entered the cell cycle but stalled in S phase. Further, we determined that cells stalled in S phase were defective in DNA synthesis and had increased levels of DNA damage, consistent with previous bulk analysis studies (*Ho and Ames, 2002*; *Ho et al., 2003*; *Yan et al., 2008*) suggesting a critical role for $Zn^{2+}$ in maintaining genome integrity during replication. Finally, we found that $Zn^{2+}$ deficiency-induced quiescence does not require p21, suggesting a mechanism distinct from spontaneous quiescence (*Spencer et al., 2013*; *Moser et al., 2018*), and follows a different pattern than mitogen withdrawal. Ultimately, our study provides new insights into when $Zn^{2+}$ is required during the mammalian cell cycle and the consequences of insufficient $Zn^{2+}$ levels.

## Results

### Nutritional $Zn^{2+}$ levels influence cell proliferation and intracellular $Zn^{2+}$ levels

To revisit the question of how $Zn^{2+}$ deficiency blocks cell growth and proliferation, we leveraged tools to visualize, track and measure molecular markers using fluorescent reporters in naturally cycling cells at the single cell level. Mammalian cells are generally recognized to contain hundreds of micromolar total $Zn^{2+}$ (*Krezel and Maret, 2006*), which they are able to concentrate from the extracellular environment. The concentration of $Zn^{2+}$ in human serum is about 12–15 µM (*Hess, 2017*) and cell culture medium typically contains 1–40 µM $Zn^{2+}$ (*Glassman et al., 1980*), much of which is supplied by the serum. To rigorously control $Zn^{2+}$ availability in our growth medium, we treated serum and insulin (major sources of $Zn^{2+}$) with Chelex 100 to scavenge $Zn^{2+}$. We then generated a minimal medium (MM) containing a low percentage of serum (1.5%) still sufficient for proliferation that contained 1.46 µM $Zn^{2+}$ as determined by Inductively Coupled Plasma Mass Spectrometry (ICP-MS), which was significantly lower than the 2.2 µM $Zn^{2+}$ measured in MM not treated with Chelex 100 (*Figure 1—figure supplement 1*). To further manipulate $Zn^{2+}$, MM was either supplemented with 30 µM $ZnCl_2$ to generate a $Zn^{2+}$ replete medium (ZR) or 2–3 µM of a $Zn^{2+}$ chelator, tris(2-pyridylmethyl)amine (TPA) to generate a $Zn^{2+}$-deficient medium (ZD) (*Huang et al., 2013*). TPA can chelate other metal ions (*Figure 1—figure supplement 2*), but with significantly lower affinity, except for Cu which binds TPA with high affinity. Using the CF4 Cu fluorescent probe (*Xiao et al., 2018*), we determined that the low concentration of TPA used in this study does not perturb the labile Cu pool. On the other hand, the Cu chelator neocuproine does deplete Cu, indicating that CF4 is capable of detecting depletion of the Cu pool in MCF10A cells (*Figure 1—figure supplement 2*). To establish the effect of ZD, MM, and ZR media on cell viability, we grew cells in the respective medium for 30 hr and measured viability using trypan blue. We also compared TPA to N,N,N',N'-tetrakis-(2-pyridylmethyl)ethylenediamine (TPEN), another $Zn^{2+}$ chelator that has been widely used in studying the effect of $Zn^{2+}$-deficiency on cell proliferation (*Beyersmann and Haase, 2001*; *Haase and Maret, 2003*; *Kaltenberg et al., 2010*). Even at the low end of TPEN concentrations

reported in the literature (3 µM), greater than 70% cell death was observed at 24 hr, compared to 3 µM TPA with ~15 % cell death. When noted, an even milder ZD condition of 2 µM TPA was used and this condition resulted in only ~1% cell death (*Figure 1—figure supplement 3*). Our results are consistent with several studies that found TPEN induces apoptosis (*Sunderman, 1995*; *Johnson et al., 2000*; *Hyun et al., 2001*; *Kolenko et al., 2001*; *Canzoniero et al., 2003*; *Corniola et al., 2008*; *Lee et al., 2008*; *Makhov et al., 2008*; *Carraway and Dobner, 2012*; *Mendivil-Perez et al., 2012*; *Zhu et al., 2017*).

To determine how defined $Zn^{2+}$ medium conditions influenced intracellular labile $Zn^{2+}$ levels in the cytosol, we created an MCF10A cell line stably expressing a genetically encoded FRET-based sensor for $Zn^{2+}$ (ZapCV2 *Fiedler et al., 2017*), grew the cells in ZD, MM, and ZR media, and measured the resting FRET ratio in individual cells. Cells grown in ZD had a significantly lower average FRET ratio than cells grown in MM, while those grown in ZR conditions had significantly higher FRET ratios (*Figure 1A*). The FRET ratio correlates with the amount of labile $Zn^{2+}$ in the cytosol, and in situ calibration suggests the respective $Zn^{2+}$ levels are approximately 1, 80, and 210 pM for ZD, MM, and ZR media, respectively (*Figure 1—figure supplement 4*), indicating that exogenous nutritional $Zn^{2+}$ levels positively influence intracellular free $Zn^{2+}$ levels.

To assess how the three nutritional $Zn^{2+}$ regimes influenced cellular proliferation, we counted cells as a function of time. Naturally cycling cells expressing H2B-mCherry were imaged for 60 hr and cells in each frame were segmented and counted using a custom automated analysis as described in *Supplementary file 1*. ZD growth conditions exhibited significantly reduced cell counts over time compared to MM and ZR conditions, with cell proliferation effectively halted after about 15 hr (*Figure 1B*). Cells grown in ZR conditions reached higher cell counts, demonstrating that increased $Zn^{2+}$ in the medium promotes cellular proliferation.

Having established that our $Zn^{2+}$-deficient conditions did not result in increased cell death, the decreased cell counts in ZD conditions could result either from a longer time between cell divisions, or an increased fraction of cells that enter a non-proliferative quiescent state. To differentiate these possibilities, we tracked individual cells over time and counted the number of mitosis events and the time between these events. Mitosis events were identified by a combination of the change in intensity of H2B-mCherry and change in size of the nucleus as described in *Supplementary file 1*. The number of mitosis events in ZD media decreased over time with few mitosis events detected after about 15 hr (*Figure 1C*). Cells grown in MM and ZR conditions underwent mitosis events throughout the observation period, with a comparable inter-mitotic time (peaking around 13 hr, *Figure 1—figure supplement 5*). The inter-mitotic time could not be measured in ZD media because few cells underwent multiple rounds of cell division. Resupply of $Zn^{2+}$ by adding back either MM or ZR media after 24 hr in ZD conditions restored cell proliferation (*Figure 1B and C*), revealing that the cells were cell-cycle competent. Combined, these results suggest that mild $Zn^{2+}$ deprivation reduces cell proliferation, not by induction of cell death, but by inducing cell cycle arrest.

## Mild $Zn^{2+}$ deficiency induces cellular quiescence and stalling of the cell cycle at an intermediate CDK2 activity

To further examine how ZD conditions halted cell division and characterize the state of cells in ZD medium, we examined single cell fate using a fluorescent reporter of CDK2 activity (*Spencer et al., 2013*). Following cell division, CDK2 activity is low and the fluorescent reporter is localized in the nucleus, but as the cell cycle proceeds CDK2 activity increases and the reporter is progressively translocated into the cytosol (*Spencer et al., 2013*). Thus, the ratio of cytosolic/nuclear fluorescence can be used as a readout for CDK2 activity and serves a 'molecular timer' for progression through the cell cycle. As described previously, CDK2 activity defines subpopulations of cells with different cell fates in a population of naturally cycling cells (*Spencer et al., 2013*; *Cappell et al., 2016*; *Moser et al., 2018*; *Arora et al., 2017*; *Yang et al., 2017*). When the CDK2 ratio remains low after mitosis (CDK2$^{low}$), a cell is classified as quiescent whereas when CDK2 activity increases above a defined threshold within 4 hr after mitosis (CDK2$^{inc}$), a cell is born committed to cell cycle entry (*Figure 2A*). A third classification has been observed, in which a cell is born with low CDK2 activity (low CDK2 ratio) but eventually ramps up activity and commits to the cell cycle (CDK2$^{emerge}$).

To define how $Zn^{2+}$ availability in the media affects cell cycle commitment, we used an MCF10A cell line stably expressing the fluorescent CDK2 reporter and H2B-mTurquoise2 and imaged cells in ZD, MM or ZR media for 60 hr. Individual cells were segmented, tracked, and analyzed for CDK2

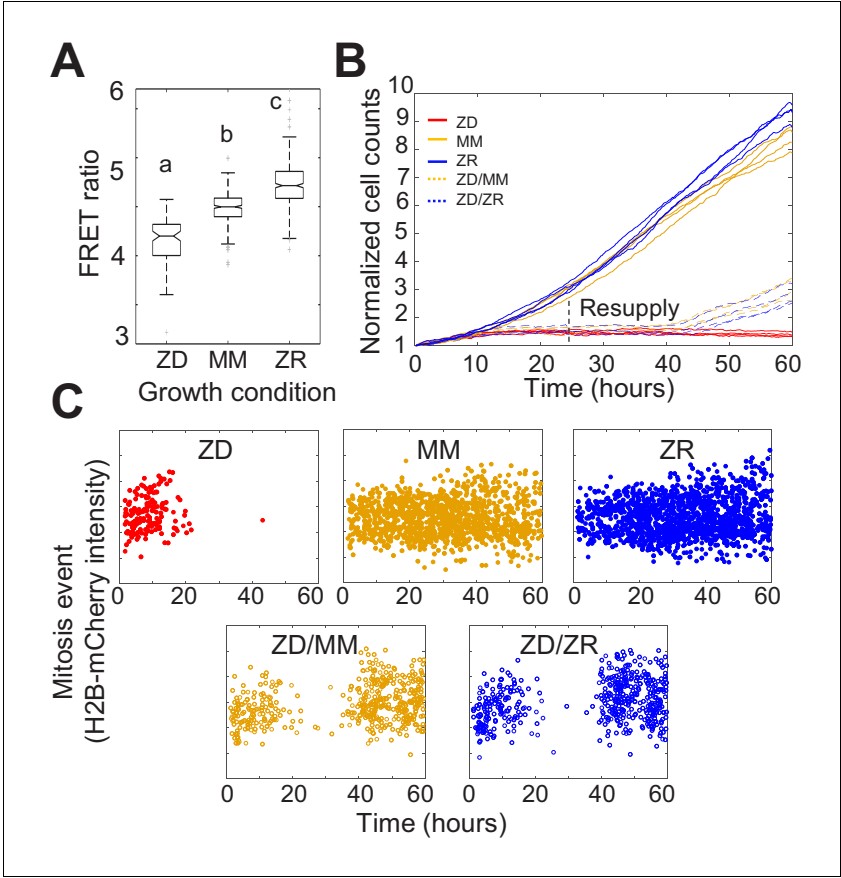

**Figure 1.** Nutritional $Zn^{2+}$ levels influence cell proliferation and intracellular $Zn^{2+}$ levels. (**A**) The FRET ratio, which is proportional to labile $Zn^{2+}$, of cells expressing ZapCV2 and grown in ZD, MM, or ZR conditions (n = 103, 382, and 402, respectively, pooled from four wells from each condition). Letters indicate statistically different groups by ANOVA with Tukey-Kramer, p<0.001 for each comparison. DF = 2 and F = 333). Average FRET ratios were calculated in a one hr window before mitosis. (**B**) Normalized cell counts after 60 hr of growth in either Zn-deficient (ZD), minimal medium (MM), or Zn replete (ZR) medium. ZD/MM and ZD/ZR conditions were grown in ZD medium until resupply with either MM or ZR medium at 24 hr. Each line represents cells grown in an individual well (n = 4 wells per condition). Cell counts were normalized to initial density in each well. (**C**) Mitosis events detected over time of cells in panel A. Mitosis events were identified as described in *Supplementary file 1* (Total mitosis events pooled from four wells).

The online version of this article includes the following source data and figure supplement(s) for figure 1:

**Source data 1.** FRET ratios of cells grown in different media.
**Figure supplement 1.** Total elemental content in minimal medium with and without Chelex 100 treatment.
**Figure supplement 1—source data 1.** ICP-MS data for total elemental content in minimal medium with and without Chelex-100.
**Figure supplement 2.** The effect of TPA on metal ions other than zinc.
**Figure supplement 3.** TPEN induces high levels of cell death, while TPA at low levels causes significantly less cell death.
**Figure supplement 3—source data 1.** Data for cell viability under different chelator concentrations.
**Figure supplement 4.** Intracellular $Zn^{2+}$ levels titrate with extracellular medium.
**Figure supplement 5.** Intermitotic time does not differ between cells grown in minimal medium and $Zn^{2+}$ replete conditions.

---

activity using a custom MATLAB pipeline (*Supplementary file 1*). In $Zn^{2+}$-sufficient medium, cells cycled naturally throughout the observation window, as evidenced by the observation of mitosis events, inter-mitotic time, and cyclical decrease in CDK2 activity after mitosis, followed by increase marking cell cycle commitment (*Figure 2B*, *Figure 1—figure supplement 5*). When cell traces from each condition were aligned computationally to mitosis, we observed all three cell fate classifications

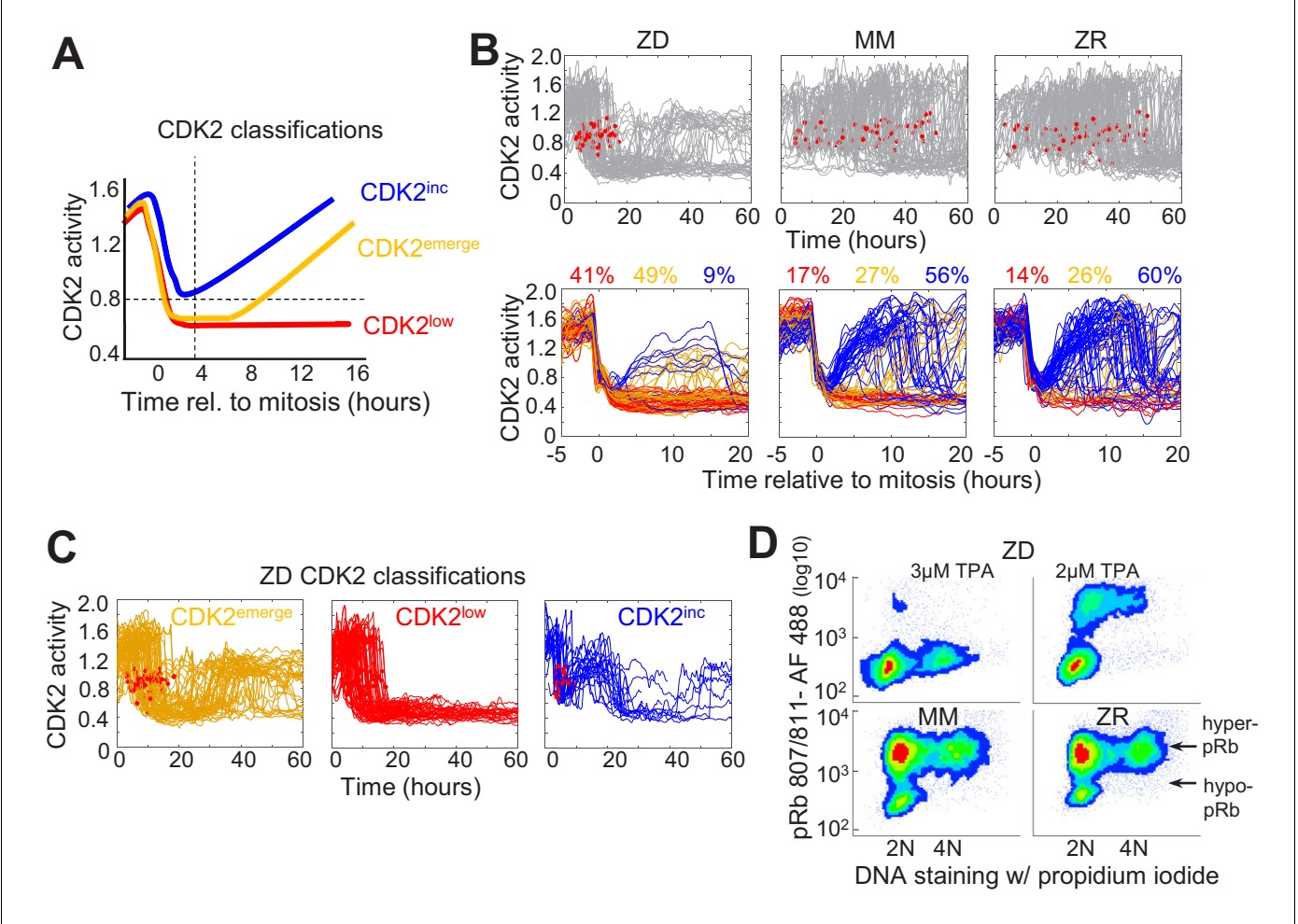

**Figure 2.** Mild Zn$^{2+}$ deficiency induces cellular quiescence and stalling of the cell cycle in S-phase. (A) Schematic of possible cell fates identified using the CDK2 activity reporter. Dashed lines indicate time and CDK2 activity thresholds used for cell fate classifications. Cell traces were computationally aligned to mitosis. (B) Single cell CDK2 traces of cells grown for 60 hr in either ZD, MM, or ZR media (n = 80 random cell traces from four individual wells). Gray CDK2 traces in top panel are displayed in real time and red dots indicate mitosis events. CDK2 traces in bottom panel are aligned computationally to mitosis and colored by cell fate classifications based on CDK2 activity after the first mitosis event. Percent of the total traces classified into each category are listed above traces (n = 254, 2015, 2059 total cells from four wells for ZD, MM, and ZR, respectively). (C) Traces from individual cell fate classifications of cells grown in ZD medium as in B (n = 80 random traces). (D) FACS analysis of cells grown for 24 hr in either ZD (3 µM or 2 µM TPA), MM, or ZR media. Density scatter plots of pRb Ser 807/811 vs. PI DNA staining are displayed for each growth condition (n = 4527, 6104, 7149, 10,010, respectively). Hyper-pRb vs. hypo-pRb populations are indicated.

The online version of this article includes the following source data and figure supplement(s) for figure 2:

**Figure supplement 1.** Total elemental content in cells grown in ZD medium compared to MM.

**Figure supplement 1—source data 1.** ICP-MS data of total elemental content in cells grown in ZD medium compared to MM.

(CDK2$^{inc}$, CDK2$^{emerge}$, CDK2$^{low}$) with a similar percentage in each category in MM and ZR media (CDK2$^{inc}$ 56% vs. 60%, CDK2$^{emerge}$ 49% vs 41%, CDK2$^{low}$ 17% vs 14% for MM and ZR, respectively). However, in ZD medium the number of mitosis events ceased after about 20 hr, the CDK2 activity either stayed low or rose to an intermediate level, and cells rarely underwent multiple rounds of the cell cycle (*Figure 2B*). Computational alignment to mitosis and cell classification revealed that few cells were born with CDK2 activity sufficient to commit to the next cell cycle (9% CDK2$^{inc}$), a significant decrease compared to MM and ZR, and there was a substantial increase in the percentage of cells with low CDK2 activity following mitosis CDK2$^{low}$ (41%). One of the most striking differences between ZD and MM or ZR conditions was the fate of cells that attempted to re-enter the cell cycle (i.e. whose CDK2 activity increased) following mitosis. As shown in *Figure 2C*, in ZD medium

CDK2^emerge cells increased CDK2 activity after a period of transient quiescence but plateaued at an intermediate CDK2 activity (a ratio of about 1.2), did not achieve maximal CDK2 activity, and failed

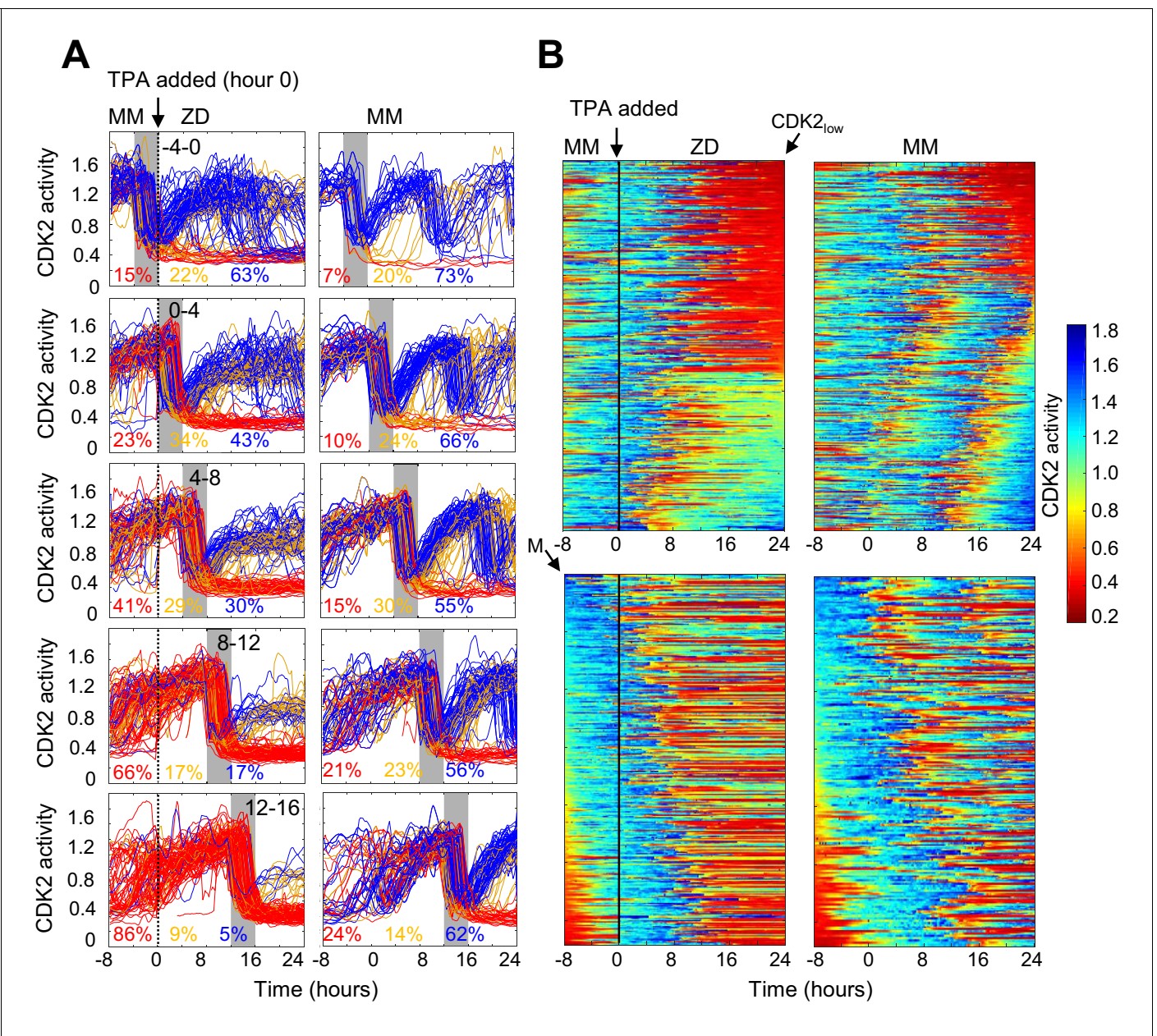

**Figure 3.** Timing of Zn$^{2+}$ deprivation with respect to cell cycle phase influences cell fate. (A) Single cell traces of CDK2 ratio over time in cells grown for 8 hr in MM followed by 24 hr in ZD conditions (left panels) or in unperturbed MM (right panels). Arrow/black vertical bar indicates time of TPA addition. Traces are separated by windows of when they underwent mitosis events relative to TPA addition. These 4 hr windows are shaded in gray (−4 to 0 hr relative to TPA addition, 0–4, 4–8, 8–12, and 12–16). Each plot shows n = 120 random traces (except the top right plot: n = 43 due to fewer cells identified in window) from 24 wells for TPA perturbed cells and eight wells for MM cells. The % traces in each category are listed below each plot. Total number of cells analyzed for these calculations: n = 237, 504, 760, 609, and 422 from top to bottom for TPA treatment windows and 43, 124, 165, 185, and 205 from top to bottom for MM windows). Cells are classified and color-coded based on the CDK2 ratio immediately following the first mitosis event. (B) Heatmaps of CDK2 traces from individual cells as grown in A, sorted computationally. Top panel heatmaps are sorted from CDK2$_{low}$ to CDK2$_{high}$ at end of the imaging period. Bottom panel heatmaps are sorted by the time of mitosis (M) at the start of the imaging period. Legend shows the CDK2 color heatmap. n = 321 traces for each heatmap.

The online version of this article includes the following source data for figure 3:

**Source data 1.** CDK2 ratios as a function of time for each data trace in *Figure 3*.

to divide. Similarly, CDK2$^{inc}$ cells increased CDK2 activity to an intermediate level before they dropped to a low level (*Figure 2C*). Analysis of mitosis events revealed that these cells did not divide before entering the CDK2$^{low}$ state. These results suggest a significant increase in the number of cells that go quiescent after mitosis and the emergence of a new cell fate, where cells attempt to re-enter the cell cycle but stall part-way through under conditions of Zn$^{2+}$ deficiency.

We also measured the elemental profile in response to 24 hr of growth in Zn-deficient conditions as compared to MM (*Figure 2—figure supplement 1*). In addition to Zn, there were significant decreases in Fe and Cu levels and significant increases in Mg, K, P, S, and Mn in ZD compared to MM growth conditions. These findings are consistent with the fact that large transcriptional changes have been identified in quiescent cells and suggest major remodeling of metal homeostasis following a period of Zn deficiency-induced quiescence.

To further define how Zn$^{2+}$ deficiency influences cell fate and characterize the consequences of the altered CDK2 activity profile in ZD cells, we examined a downstream CDK2 substrate, retinoblastoma protein (Rb). When CDK2 levels are low, Rb binds to and inhibits E2F family transcription factors, blocking cell cycle progression (*Giacinti and Giordano, 2006*). As CDK2 activity increases, Rb gets hyper-phosphorylated which releases the inhibition, enabling E2F to transcribe cell cycle genes. A previous study showed that cells born with elevated CDK2 activity also had hyper-phosphorylated Rb (pRb), as determined by immunofluorescence, whereas cells born with CDK2$^{low}$ had low levels of phosphorylated Rb (*Spencer et al., 2013*). We employed a similar protocol to measure phosphorylated Rb and DNA content by Fluorescence Activated Cell Sorting (FACS). After 24 hr of growth in MM or ZR media, the majority of cells had hyper-pRb with either 2N, intermediate, or 4N DNA content, suggesting cells were actively cycling through G1, S, G2, and M (*Figure 2D*). The small fraction of cells with hypo-pRb and 2N DNA content, correspond to the small fraction of cells with CDK2$^{low}$ and represent quiescent cells. Treatment of cells with 2 µM TPA for 24 hr revealed that most cells had 2N DNA content and hypo-pRb, consistent with most cells being in a quiescent state. However, some cells had hyper-pRb, indicating elevated CDK2 activity and an attempt to progress through the cell cycle, although there was a decrease in the fraction of cells with 4N DNA content indicating a deficiency in DNA replication. With 3 µM TPA the majority of cells had hypo-pRb with 2N DNA content, consistent with a quiescent state. A small population of cells had hypo-pRb and 4N DNA content, suggesting that after DNA replication, the cells entered quiescence without undergoing mitosis, consistent with the CDK2$^{inc}$ population of cells in *Figure 2C* that slips back to a CDK2$^{low}$ state. Combined, these results indicate that Zn$^{2+}$ is required for cell cycle progression and there is heterogeneity in the cellular response to Zn$^{2+}$ deprivation; some cells are born with low CDK2 activity and immediately enter quiescence, while others are born with elevated CDK2 activity (CDK2$^{inc}$) or increase CDK2 activity after some delay (CDK2$^{emerge}$). Further, our results suggest that the milder the Zn$^{2+}$ deficiency, the more cells attempt to progress through the next cell cycle following mitosis. However, there is a clear requirement for Zn$^{2+}$ to successfully progress past S-phase to G2/M, which we explore below.

## Timing of Zn$^{2+}$ removal with respect to the cell cycle state influences cell fate

The experiments in *Figure 2* revealed heterogeneity in cell fate in response to Zn$^{2+}$ deficiency. Given that the cells were cycling asynchronously prior to Zn$^{2+}$ deprivation, we wondered whether the cell fate was determined by a cell's position in the cell cycle at the time of Zn$^{2+}$ withdrawal. To address this, we imaged cells expressing the CDK2 sensor in MM for 8 hr to track cell cycle progression prior to Zn$^{2+}$ deprivation and follow entry of cells into quiescence. We binned cell traces according to when cells divided within specific 4 hr windows relative to TPA addition (hr 0), from 4 hr before TPA addition (−4 to 0) up to 16 hr after TPA addition (*Figure 3A*, gray shaded boxes). For cells that divided within 4 hr prior to TPA addition, a small but elevated percent of cells went quiescent (15% vs. 7% in MM), suggesting that cells need Zn$^{2+}$ when exiting mitosis and progressing into G1. Still, when cells divided within 4 hr of Zn$^{2+}$ deprivation, the majority of cells re-enter the cell cycle either immediately (CDK2$^{inc}$) or with a slight delay (CDK2$^{emerge}$). What was striking about this population of cells, was that only a small fraction was able to complete the next round of cell division compared to cells in MM conditions (*Figure 3A* top two panels), and instead most cells stalled with an intermediate CDK2 ratio. Thus, even if cells are born with elevated CDK2 activity and pass the classical restriction point defined by a need for extracellular growth signals such as mitogens (*Pardee, 1974*), they

rarely progress past an intermediate CDK2 activity in the absence of $Zn^{2+}$. In cells that divided in subsequent windows of time after TPA addition, there was a progressive decrease in the percentage of cells born with elevated CDK2 activity and classified as CDK2$^{inc}$ (43%, 30%, 17%, and 5%), indicating that longer $Zn^{2+}$ deprivation increases the probability of cells entering quiescence after mitosis (*Figure 3A* top to bottom). There was a small increase in the percent of quiescent cells in unperturbed MM over time due to increased cell density and quiescence induced by contact inhibition. Notably, when $Zn^{2+}$ was removed prior to cell division and cells attempted to enter another round of the cell cycle, they stalled at an intermediate CDK2 activity, consistent with an inability to progress past S-phase in the absence of $Zn^{2+}$.

To more precisely examine the timing of events ($Zn^{2+}$ removal, mitosis and cell fate) and compare to the behavior to mitogen removal, we plotted heat maps of individual cell traces over time (*Figure 3B*). We computationally grouped cell traces by their CDK2 activity at the end of the growth period, demonstrating 1) the two major cell fates for ZD cells: quiescence in red and stalled at intermediate CDK2 in green-turquoise and 2) that if cells divided more than 8 hr after TPA addition they entered quiescence immediately after cell division (*Figure 3B* top panel). If cells divided within 8 hr of TPA addition, the majority re-entered the cell cycle and stalled at an intermediate CDK2 activity (green-turquoise), consistent with an inability to progress past S-phase. As the time between TPA addition and cell division increased from 0 to ~8 hr, a greater proportion of cells experienced a prolonged low CDK2 activity period (longer red streak) before re-entering the cell cycle. These results suggest that if cells experience a short window of $Zn^{2+}$ deficiency they have a greater chance of re-entering the cell cycle; but as the length of time increases, cells experience prolonged bouts of low CDK2 activity, suggesting that $Zn^{2+}$ plays a role in processes involved in ramping up CDK2 activity to promote cell cycle entry. In contrast, the majority of cells in MM continued to cycle throughout the measurement window.

To compare $Zn^{2+}$ deficiency-induced quiescence to mitogen withdrawal, we aligned CKD2 traces computationally to the time of mitosis, with the first mitosis event at the top of the heat map (3B, bottom panel). Previously, this analysis demonstrated that if mitogens are withdrawn from newly born CDK2$^{inc}$ cells, they completed one additional round of the cell cycle (*Spencer et al., 2013*), indicating that achieving a certain threshold of CDK2 activity marks cell-cycle commitment, regardless of mitogen availability. When aligned in a similar manner (*Figure 3B*, bottom), our traces reveal that the CDK2 activity window that defines cell cycle commitment with respect to mitogen removal does not apply to $Zn^{2+}$ removal. Instead, as other data representations suggest, even if cells pass the restriction point for mitogens, they stall at an intermediate CDK2 activity in the absence of sufficient $Zn^{2+}$, suggesting that $Zn^{2+}$ deficiency-induced quiescence acts through a different pathway compared to mitogen withdrawal.

## $Zn^{2+}$ deficiency causes a defect in DNA synthesis

Because we found a large population of cells which stalled at an intermediate CDK2 activity under $Zn^{2+}$-deficient conditions and speculated that these cells were stalled in S phase, we wanted to measure whether these cells were capable of DNA synthesis. We grew cells as in *Figure 3*, measured 5-ethynyl-2'-deoxyuridine (EdU) incorporation during a 15 min window of labeling at the end of this growth period, and stained for DNA content using propidium iodide/RNAse. Plotting EdU intensity against DNA content for single cells revealed the expected distribution of cell cycle phases, where EdU negative cells (Edu⁻) with 2N DNA were in quiescence (G0) or G1, EdU⁻ cells with 4N DNA were in G2 or M, and EdU positive cells (Edu⁺) transitioning between 2N and 4N were in S phase (*Figure 4A*, cell cycle phases shown in boxes on right plot). In MM and ZR, the Edu vs. DNA content density plot followed a classical arch distribution, as has been found previously for MCF10A in full growth medium (*Gookin et al., 2017*). In ZD conditions, cells did not exhibit high EdU intensity, indicating normal DNA synthesis was impaired, and a large portion of cells were EdU⁻ with 2N DNA content, consistent with quiescence. Interestingly, in ZD conditions, many cells between 2N and 4N DNA content exhibited some Edu staining above that of cells classified as EdU⁻, suggesting that some cells are able to enter S phase, begin DNA synthesis, but at a reduced rate in the 15 min interval. The EdU intensities for cells grown in 2 μM TPA were slightly higher than those for cells grown in 3 μM TPA, demonstrating that 2 μM TPA is a milder ZD condition and impairment of DNA synthesis was less severe. These data confirm our findings from *Figure 3*, where in addition to quiescence, a cell fate with intermediate CDK2 activity exists. This state of intermediate CDK2 activity is indeed

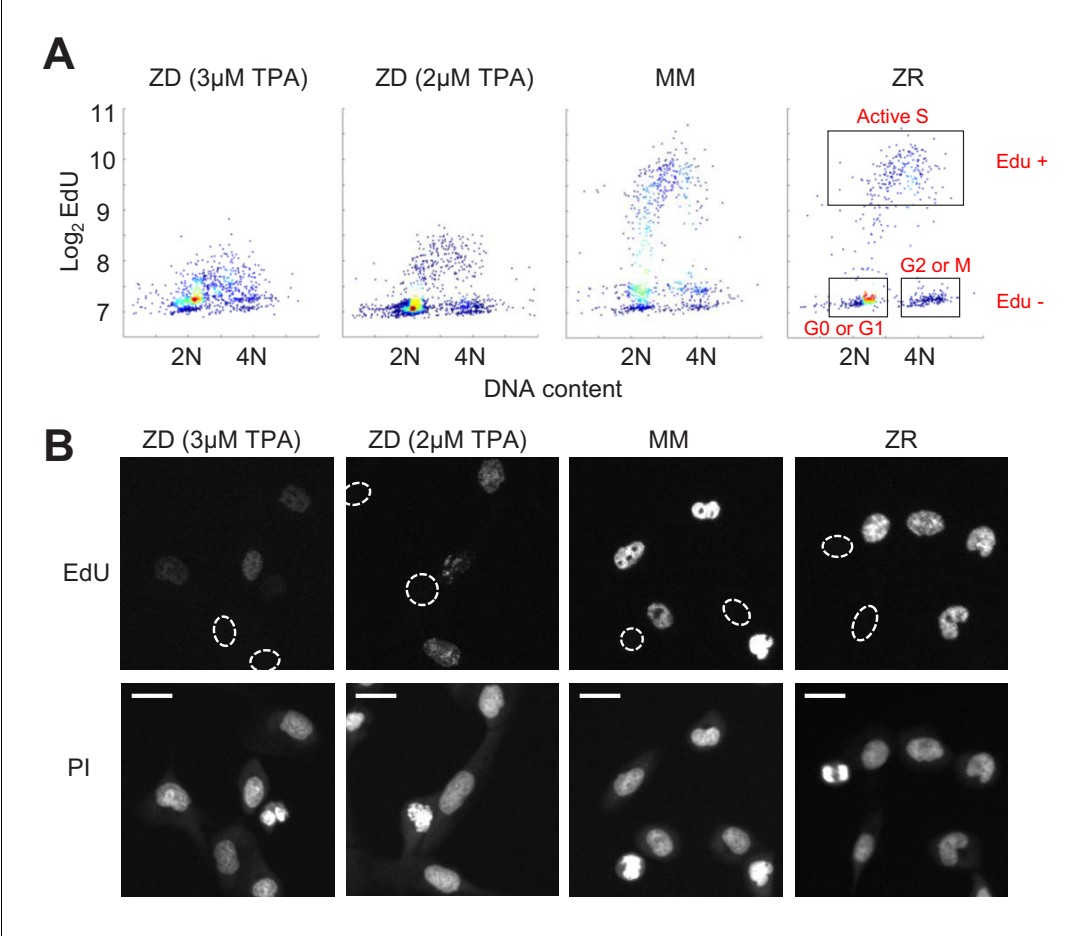

**Figure 4.** Zn[2+] deficiency causes a defect in DNA synthesis. (**A**) Density plots of single cells analyzed for EdU incorporation and DNA content. EdU incorporation was measured after 15 mins of incubation after 24 hr of growth in either ZD (3 μM or 2 μM TPA), MM, or ZR conditions (n = 1437 pooled from two wells, 916 from two wells, 817 from four wells, 676 from four wells, respectively, by condition). Propidium iodide (PI) plus RNAse was used to stain for DNA content define 2N and 4N populations. Cell cycle phases were classified by their location on the 2D Edu vs. DNA plot (shown in red). (**B**) Representative cell images for Edu and PI staining for each growth condition. EdU negative cells are outlined with a white dash lined. Scale bars indicate 20 μm.

S phase, as indicated by cells undergoing DNA synthesis, albeit at a reduced rate. Thus, though these cells have crossed a G1/S transition, Zn[2+] is required in S phase for DNA synthesis to proceed at a normal rate and for cells to complete DNA synthesis and progress to G2/M.

## Zn[2+] deficiency leads to increased levels of DNA damage in cycling cells

Cells that experience mild DNA damage can temporarily exit the cell cycle and enter a quiescent state in order to avoid passing a damaged genome on to the next generation. Given that long-term (multi-day) growth in Zn[2+]-deficient medium has previously been shown to increase DNA damage in a population of cells by the comet assay (*Ho and Ames, 2002*; *Ho et al., 2003*; *Yan et al., 2008*), we wondered whether mild Zn[2+] deficiency could induce DNA damage on a shorter timescale and whether this DNA damage could be a cause of quiescence induced by Zn[2+] deficiency. Further, because we found that ZD conditions caused a defect in DNA synthesis, we wanted to assess whether cells stalled in S phase were experiencing replication stress. Because we observed heterogeneity in cell cycle fates upon Zn[2+] withdrawal, we sought to measure DNA damage at the single cell level and correlate it with a cell cycle marker. Previously, Arora and coworkers determined that whether naturally cycling cells enter quiescence and how long they spend in quiescence is in part explained by levels of double-stranded break (DSB) DNA damage inherited from mother cells, as measured by tracking fluorescent 53BP1 foci (a known marker of DSBs) (*Arora et al., 2017*). Around

the same time, Barr and coworkers demonstrated that both DSBs and single strand breaks (SSBs, measured by RPA2 foci) during S phase contribute to induction of quiescence (*Barr et al., 2017*). We used a similar approach, quantifying 53BP1 and RPA2 puncta in individual cells to identify the presence of DSB and SSB, respectively. We correlated DNA damage markers with the cell cycle using phospho-Rb status in thousands of individual cells exposed to 24 hr of either ZD, MM, or ZR growth conditions, where cells that were hypo-pRb were classified as quiescent, while hyper-pRb cells were classified as cycling (G1, S, G2 or M). For cells classified as quiescent (hypo-pRb), there was no significant difference in the fraction of cells with 53BP1 foci (DSBs) and a small but statistically significant increase in RPA2 foci (SSB) (4% versus <1% in MM and ZR) in ZD versus MM and ZR media. These results suggest that DNA damage is likely not the primary mechanism of quiescence induction in $Zn^{2+}$-deficient cells. However, in cycling cells, there was a significant increase in 53BP1 foci (87% in ZD versus ~45% in MM and 41 ZR) and RPA2 foci (~17% in ZD versus <1% in MM and ZR). DNA damage was measured after 24 hr in the respective medium, and for cells in ZD, this time point corresponds to about 60% of the cells in a low CDK2 activity/quiescent state and 40% of the cells stalled at a state with intermediate CDK2 activity, consistent with S-phase (*Figures 3B*, 24 hr time point). Given the increase in DNA damage in cycling cells, but relatively subtle change in DNA damage in quiescent cells, we speculate that the cells with increased DNA damage correspond to the cells stalled in the cell cycle at S phase (*Figure 3*), and that $Zn^{2+}$ deficiency induces a defect in DNA synthesis (*Figure 4*) that contributes to the inability of these cells to progress to G2/M. Our results also suggest that $Zn^{2+}$ deficiency can induce quiescence independent of induction of DNA damage because those cells that have gone quiescent by 24 hr do not exhibit a profound increase in DNA damage.

## Quiescence induced by $Zn^{2+}$ deficiency does not require p21

p21 is a cyclin dependent kinase inhibitor that binds to and inhibits the activity of cyclin-CDK complexes, thus regulating cell cycle progression. p21 is upregulated in response to contact inhibition and growth factor withdrawal and contributes to cell cycle arrest upon these perturbations (*Perucca et al., 2009*). Furthermore, p21 is a transcriptional target of p53 and has been shown to be upregulated in response to DNA damage, which results in cell cycle arrest and presumably enables DNA repair prior to cell cycle re-entry. Thus, p21 has emerged as an important regulator of the proliferation-quiescence decision (*Moser et al., 2018*; *Barr et al., 2017*). This is underscored by the observation that in p21 null cells, the incidence of spontaneous quiescence is reduced (*Spencer et al., 2013*; *Arora et al., 2017*). Given that $Zn^{2+}$ deficiency influences cell cycle progression by inducing quiescence and that it also results in increased DNA damage, we sought to determine whether quiescence resulting from $Zn^{2+}$ deficiency requires p21. We grew WT and p21$^{-/-}$ MCF10A cells expressing the CDK2 reporter and measured the fraction of cells in each classification (CDK2$^{low/emerge/high}$) under ZD, MM, and ZR conditions (Figure 6A). p21 knockdown was validated as described in the methods section (*Figure 6—figure supplement 1*). In WT cells grown in MM or ZR, 17% or 16% (respectively) of cells were classified as quiescent (CDK2$^{low}$), while in p21$^{-/-}$ cells, only 8% or 5% cells were classified as quiescent, consistent with previous results suggesting that spontaneous quiescence in naturally cycling cells is induced by endogenous DNA damage and is dependent on p21 (*Arora et al., 2017*; *Barr et al., 2017*). In ZD conditions quiescence occurred at a similar rate in p21$^{-/-}$ and WT cells (42% vs. 56%), suggesting that quiescence caused by $Zn^{2+}$ deficiency does not explicitly require p21 (*Figure 6A*). This is perhaps not surprising, given that the cell population that was quiescent after 24 hr of growth in $Zn^{2+}$-deficient medium did not exhibit increased DNA damage (*Figure 5*). Combined, our data indicate that $Zn^{2+}$-deficiency induces quiescence via a p21-independent pathway. Because p21 is also upregulated to maintain quiescence (*Coller et al., 2006*), we measured p21 levels in WT MCF10A cells using immunofluorescence after 40 hr of growth. In ZD medium, cells that maintained low CDK2 activity had higher levels of p21, similar to cells in MM medium (*Figure 6B*). These data suggest that although p21 is not required for entry into quiescence, in WT cells p21 is upregulated when cells are quiescent, likely to maintain their quiescent state by suppressing CDK2 activity.

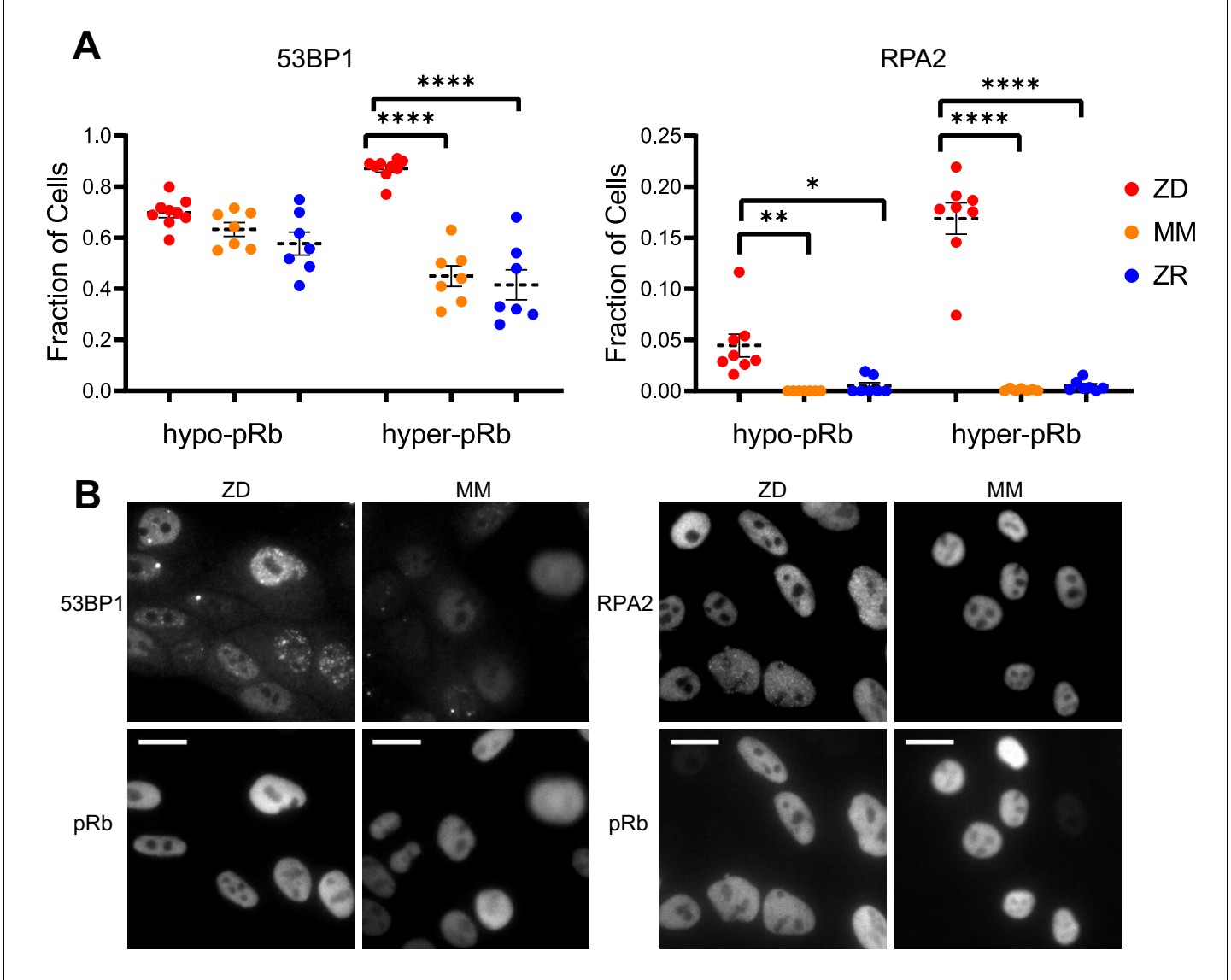

**Figure 5.** $Zn^{2+}$ deficiency leads to increased DNA damage. (**A**) Cells were grown in ZD, MM, or ZR media for 24 hr and then stained with antibodies for pRb and either 53BP1 (a marker of DSB) or RPA2 (a marker of SSB). The pRb status of each cell was used to classify cells as either quiescent (hypo-pRb) or cycling (hyper-pRb). Foci were identified using a custom MATLAB script. The fraction of cells positive for DNA damage (presence of 1+ foci) is plotted. Each point represents an individual well from two pooled biological replicate experiments (n wells = 6–9 per condition).The total n values for each condition are as follows: 53BP1: 5940, 6069, and 7473 for ZD, MM, and ZR; RPA2: 4454, 5991, and 6480 for ZD, MM, and ZR. Statistical significance was assessed using one-way ANOVA with Sidak's multiple corrections test (*p<0.05, **p<0.01, ****p<0.0001). Exact p-values for 53BP1: hyper-ZD vs hyper-MM = 5.54E-10, hyper-ZD vs hyper-ZR = 6.61E-11; Exact p-values for RPA2: hypo-ZD vs hypo-MM = 4.17E-03; hypo-ZD vs hypo-ZR = 1.35E-02, hyper-ZD vs hyper-MM = 1.00E-15, hyper-ZD vs hyper-ZR = 3.00E-15. (**B**) Representative images for 53BP1 and RPA2 staining for ZD and MM conditions. Scale bars indicate 20 μm.

The online version of this article includes the following source data for figure 5:

**Source data 1.** Cell parameters for DNA damage assay.

## $Zn^{2+}$ resupply induces synchronized cell cycle re-entry

To determine whether $Zn^{2+}$ deficiency-induced quiescence is reversible, we grew cells for 24 hr in ZD medium followed by 36 hr with either MM or ZR media. $Zn^{2+}$ resupply, by adding either MM or ZR media, caused CDK2 activity to increase and the resumption of mitosis events (red dots), indicating active cell division (*Figure 7A*). Notably, when individual elements contained in MM were resupplied, only $Zn^{2+}$, but not $Fe^{2+}$ or $Cu^{2+}$ rescued ZD-induced quiescence, suggesting $Zn^{2+}$ alone is

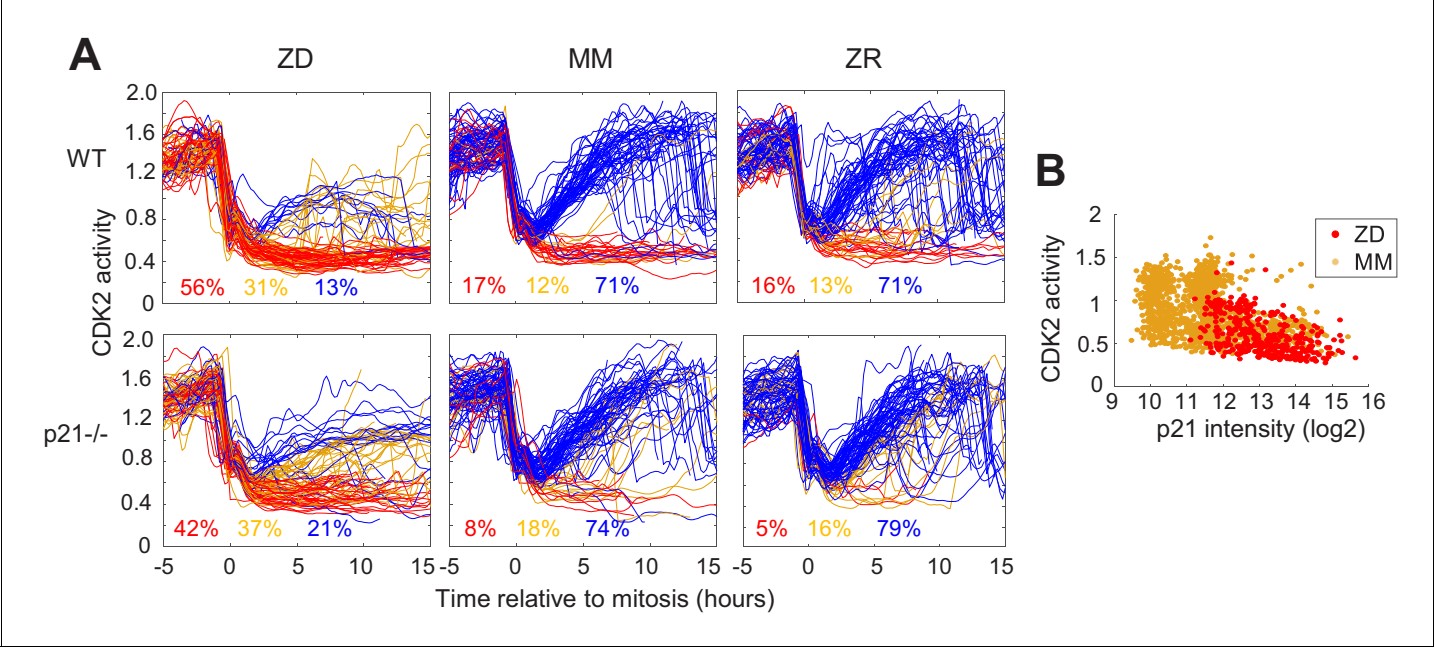

**Figure 6.** Quiescence induced by Zn²⁺ deficiency does not require p21. (**A**) Single cell CDK2 traces of WT (top) and p21⁻/⁻ (bottom) MCF10A cells grown in either ZD, MM, or ZR media. Plots show n = 120 random traces selected from eight individual wells. CDK2 traces were aligned computationally to mitosis and colored by cell fate classifications. % total traces classified into each category are listed below traces (calculated from total traces n = 410, 3275, 3445 for WT cells grown in ZD, MM, and ZR, respectively and n = 441, 764, and 659 for p21⁻/⁻ cells). (**B**) CDK2 ratio vs. p21 intensity in WT MCF10A cells at a fixed timepoint after 40 hr of growth in ZD or MM media. p21 was detected with an anti-p21 antibody (for ZD, n = 412 cells pooled from three wells; for MM, n = 1414 cells pooled from three wells).

The online version of this article includes the following figure supplement(s) for figure 6:

**Figure supplement 1.** Validation of p21 knockdown in p21⁻/⁻ cell line.

sufficient to induce cell-cycle re-entry, despite the reduction in Fe and Cu measured in quiescent cells by ICP-MS (*Figure 7—figure supplement 1*, *Figure 2—figure supplement 1*). It appeared that a smaller subset of cells remained quiescent when rescued with ZR as opposed to when rescued with MM (see CDK2ˡᵒʷ traces in highlighted windows in *Figure 7A*). To quantify this, we generated CDK2 activity probability density plots for each hr after resupply with either MM or ZR (*Figure 7B* and *Figure 7—figure supplement 1*). After 1 hr of Zn²⁺ resupply, the majority of cells were quiescent in all three conditions (CDK2ˡᵒʷ, activity mean ~0.5). After 8 hr of resupply, cells not resupplied (ZD) remained in a CDK2ˡᵒʷ state, while resupplied cells emerged from quiescence and entered the cell cycle, as indicated by the cell populations shifting towards higher CDK2 activity, with a mean around 1.25. The ZR resupplied cells had a higher probability of being in this higher CDK2 state and a corresponding lower probability of being in the CDK2ˡᵒʷ state compared to cells resupplied with MM, suggesting a positive correlation between the amount of Zn²⁺ in the medium and cell cycle re-entry after a period of deficiency.

## Discussion

Accurate duplication of the genome and separation of chromosomes into daughter cells through the process of the cell-division cycle is one of the most essential functions individual cells must execute. Regulated exit of the cell cycle to quiescence is an important quality control pathway that reduces metabolic and biochemical activities and protects cells against stress and toxic metabolites (*Tümpel and Rudolph, 2019*). Given the importance of understanding the factors that regulate cell cycle entry, progression, and exit to quiescence, decades of research have sought to define the underlying mechanisms of the proliferation-quiescence 'decision'. Much of our understanding of the mammalian cell cycle has derived from studies in which populations of cells were forced to exit the cell cycle upon induction of stress, such as serum starvation or amino acid deprivation. These studies

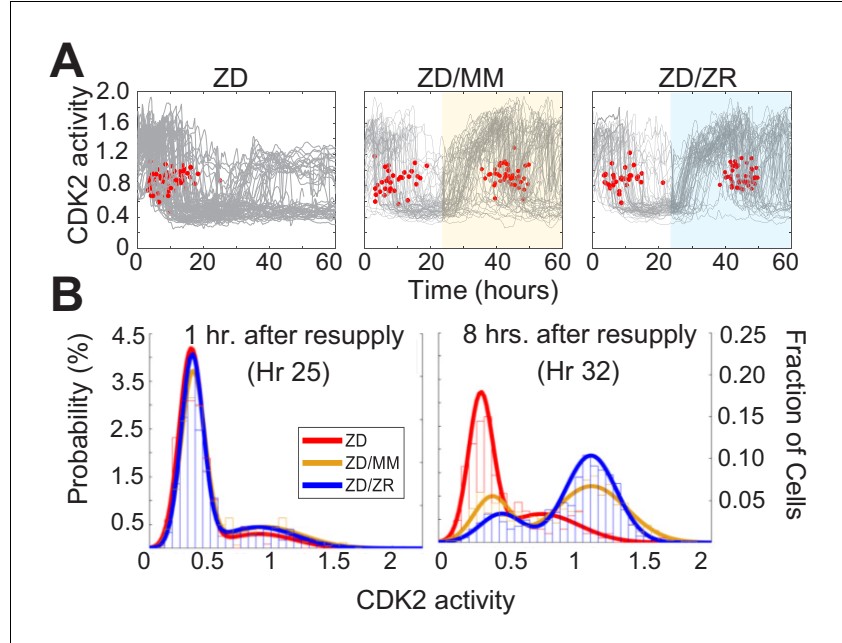

**Figure 7.** Zn²⁺ resupply after Zn²⁺-deficiency induced quiescence allows re-entry into the cell cycle. (**A**) Single cell CDK2 traces of cells grown in ZD conditions for 60 hr and cells grown in ZD for 24 hr and resupplied with either MM or ZR media (n = 80 random traces from four wells per condition). Yellow or blue shaded regions indicate medium resupply. Red dots indicate mitosis events. (**B**) CDK2 ratio density histograms after cells either remain in ZD medium or are resupplied with MM or ZR. Histograms are shown for timepoints 1 and 8 hr after resupply (n = 210, 1068, and 1024 for ZD, MM, and ZR respectively for hr 1; n = 187, 1154, and 1139 for ZD, MM, and ZR respectively for hr 8, pooled from four wells for ZD and eight wells for ZD/MM and ZD/ZR).

The online version of this article includes the following figure supplement(s) for figure 7:

**Figure supplement 1.** Zn²⁺ resupply allows for re-entry into the cell cycle following Zn²⁺ deficiency-induced quiescence.

**Figure supplement 2.** Zn²⁺ resupply, but not Cu²⁺ or Fe²⁺ allows for re-entry into the cell cycle following Zn²⁺ deficiency-induced quiescence.

---

have revealed many of the key regulators of the cell cycle and introduced the concept of a restriction point, a point at which cells commit to the cell cycle, and become mitogen-independent. However, recent application of single cell technologies for tracking cellular and molecular markers and the fate of individual cells have led to key revisions of the textbook model of the mammalian cell cycle, including the discovery that in naturally cycling cells there are multiple proliferation decisions (*Matson and Cook, 2017*; *Spencer et al., 2013*; *Cappell et al., 2016*; *Heldt et al., 2018*). Further-more, it is now appreciated, but still poorly understood that quiescence is not a single dormant state, but rather an assemblage of heterogeneous states, that is actively maintained (*Matson and Cook, 2017*; *Coller et al., 2006*; *Yao, 2014*). Broadly speaking, control of the decision to proliferate or arrest is fundamental to various aspects of tissue architecture maintenance, differentiation, DNA damage repair, wound healing, and normal vs. cancerous cell growth (*Matson and Cook, 2017*; *Heldt et al., 2018*; *Hanahan and Weinberg, 2011*). Thus, understanding how individual triggers act to induce quiescence is important for ultimately identifying targets for improving nutritional deficiencies or disease states.

Building on the emerging conceptual framework of studying the cell cycle in asynchronous populations of cells, in this work we apply a combination of fluorescent reporters, high throughput imaging, and quantitative image analysis to revisit the important but unresolved question of how Zn²⁺ deficiency blocks cell proliferation. While it has long been recognized that Zn²⁺ is required for cell proliferation, how Zn²⁺ deficiency influences the cell cycle on a single cell level has not been eluci-dated and whether Zn²⁺ affects the critical proliferation-quiescence cell fate decision has not been determined. Zn²⁺ is an essential metal that serves as a critical cofactor in approximately 10% of the

proteins encoded by the human genome (*Andreini et al., 2006*), including over 700 $Zn^{2+}$-finger containing transcription factors, DNA polymerase, superoxide dismutase, and proteins involved in DNA repair (*Lambert et al., 2018*). Thus, it is required for several key cellular processes involving these proteins, including processes relevant for the cell cycle such as transcription, antioxidant defense, DNA synthesis and repair.

It is often overlooked that in addition to serving as a reservoir for mitogens, serum is also the major source of essential micronutrients including Zn, Fe, Cu, and Mn, and thus complete removal of serum also eliminates exogenous supply of these and other essential nutrients. We sought to isolate the effect of $Zn^{2+}$ by examining $Zn^{2+}$ deprivation in an asynchronous population of cells still containing sufficient mitogen levels. We subjected cells to mild $Zn^{2+}$ perturbations that altered the labile pool of cytosolic $Zn^{2+}$ between 1 and 210 pM and avoided induction of cell death. Upon $Zn^{2+}$ deprivation, cells lost the ability to proliferate and the majority experienced one of two cell fates: entry into quiescence or stall in S phase. This result indicates that unlike mitogens there is not a single restriction point for $Zn^{2+}$, perhaps because it would be too hard to evolve a checkpoint for every nutrient, or perhaps because $Zn^{2+}$ is essential for so many cellular processes. Tracking individual cells before and after $Zn^{2+}$ deprivation revealed the temporal development of these two distinct fates, demonstrating that $Zn^{2+}$ is required at multiple places in the cell cycle and the depth of $Zn^{2+}$ deficiency relative to mitosis determines the cell fate. If cells underwent mitosis within 8 hr of $Zn^{2+}$ withdrawal, they were likely to re-enter the cell cycle and complete G1 before stalling in S phase, with a markedly prolonged S-phase (plateau at intermediate CDK2 activity in *Figure 3A*). These cells synthesized DNA at reduced rates and accumulated increased amounts of DNA damage, indicating that sufficient $Zn^{2+}$ is necessary for maintaining DNA integrity. These results are consistent with previous work which showed at a population level that long term (multi-day) $Zn^{2+}$ deficiency led to increased DNA damage as measured by a comet assay (*Ho and Ames, 2002*; *Ho et al., 2003*). Here, we find that $Zn^{2+}$ deficiency increases both DSBs and SSBs in cycling hyper-phosphorylated Rb cells (~1.9 fold increase in DSBs and ~170 fold increase in SSBs compared to cells in MM or ZR). It is well established that $Zn^{2+}$ is a critical cofactor in a number of genome caretaker proteins, such as XPA and RPA of the nucleotide excision repair pathway, PARP involved in base excision repair, HERC2 involved in DSB recognition, CHFR involved in checkpoint regulation, and transcription factors such as p53 and BRCA that are involved in DNA damage response and repair (*Danielsen et al., 2012*; *Ahel et al., 2008*; *Witkiewicz-Kucharczyk and Bal, 2006*). While in vitro studies indicate these proteins would not function without $Zn^{2+}$, little is known about the metal-binding properties in cells, how the myriad proteins of the zinc proteome acquire their $Zn^{2+}$, and whether these proteins are sensitive to subtle perturbations of the labile zinc pool. Ho and coworkers demonstrated that long-term $Zn^{2+}$ depletion decreased the binding of transcription factors such as p53 and AP1 to DNA (*Ho and Ames, 2002*), and resulted in differential expression of genes involved in the cell cycle as well as DNA damage response and repair (*Yan et al., 2008*). However, nothing was known about how rapidly cells sense $Zn^{2+}$ deficiency and how consequences of $Zn^{2+}$ deficiency, such as increased DNA damage, correlate with cell fate. By following the fate of cells over time in response to $Zn^{2+}$ withdrawal, we show that individual cells sense depletion of the labile $Zn^{2+}$ pool quickly (within 4 hr), and that cells in S-phase experience decreased rates of DNA synthesis and increased DNA damage, inducing cell cycle exit, despite these cells having sufficient mitogens to have passed the restriction point.

If cells divided >8 hr after $Zn^{2+}$ withdrawal, the vast majority were born with low CDK2 activity and immediately entered a quiescent state. Intriguingly, these cells didn't exhibit substantial increases in DNA damage and p21 was not required for entry into this quiescent state, indicating the mechanism of quiescence-induction is distinct from that of spontaneous quiescence. Entry into 'spontaneous' quiescence observed in naturally cycling cells was found to be associated with increased levels of DNA damage from the previous cell cycle was dependent on p21 activity (*Arora et al., 2017*; *Barr et al., 2017*). The observation that cells were born with low CDK2 activity suggests that $Zn^{2+}$ deficiency is sensed in the mother cell during the previous cell cycle. This new micronutrient deficiency-induced quiescent state occurs in the presence of mitogens and is not induced via replication stress/DNA damage. Importantly, quiescent cells and cells stalled in S phase after a period of $Zn^{2+}$ deficiency could be rescued by resupply of $Zn^{2+}$ and were competent to re-enter the cell cycle. Interestingly, we found that while the TPA chelator didn't perturb the labile pool of other metals such as Cu, the $Zn^{2+}$-deficient quiescent state was characterized by notable changes

in other essential metals, such as Cu, Fe, and Mn, suggesting significant changes to metal homeostasis in this quiescent state. Although quiescence was rescued exclusively by Zn (and not by Cu or Fe), it would be intriguing to study the mechanisms of metal homeostasis remodelling in a future study.

By addressing the question of how mild $Zn^{2+}$ deficiency reduces cell proliferation with modern tools at the single cell level in naturally cycling cells, we revealed that $Zn^{2+}$ status influences multiple checkpoints in the cell cycle, and uncovered a new quiescent state induced by micronutrient deficiency that is distinct from spontaneous quiescence and mitogen withdrawal. These findings are important for understanding how this cell cycle control might be perturbed in disease states such as cancer where $Zn^{2+}$ levels and localization have been shown to be perturbed (*Hanahan and Weinberg, 2011*; *Prasad et al., 2009*; *Pan, 2017*; *Chandler et al., 2016*). Future proteomic and transcriptomic studies have the potential to reveal the relevant $Zn^{2+}$ dependent proteins that sense $Zn^{2+}$ status, thus uncovering the mechanism of entry into this quiescent state and the key $Zn^{2+}$-dependent proteins responsible for maintaining genome integrity during DNA replication.

# Materials and methods

## Key resources table

| Reagent type (species) or resource | Designation | Source or reference | Identifiers | Additional information |
|---|---|---|---|---|
| Cell line (*Homo sapiens*) | MCF10A | ATCC | CRL-10317, RRID:CVCL_0598 | |
| Cell line (*Homo sapiens*) | MCF10A + PB-NES ZapCV2 and PB-H2B- mCherry (stable) | This paper | | |
| Cell line (*Homo sapiens*) | MCF10A + DHB-Venus and H2B-mTurqoise (stable) | Laboratory of Sabrina Spencer | PMCID: PMC4001917 | |
| Cell line (*Homo sapiens*) | MCF10A p21$^{-/-}$ + DHB-Venus and H2B-m Turqoise (stable) | Laboratory of Sabrina Spencer | PMCID: PMC4001917 | |
| Recombinant DNA reagent | PB-NES-ZapCV2 (plasmid used for stable cell line generation) | Published in previous Palmer lab paper PMID: 27959493 | | Cytosolic FRET sensor in Piggybac vector |
| Recombinant DNA reagent | PB-H2B-mCherry (plasmid used for stable cell line generation) | This paper | | H2B-mCherry subcloned from H2B-mCherry CSII-EF (PMCID:PMC4001917) into PB510B1 Piggybac vector (System Biosciences) with primers listed below |
| Sequence-based reagent | Forward primer to subclone H2B-mCherry into PB plasmid (PB510B1, System Biosciences) | IDT | | 5'-GGAATT GAATTCGCT ACCGGTCTC GAGAC-3' (EcoRI site for cloning) |
| Sequence-based reagent | Reverse primer to subclone H2B-mCherry into PB plasmid (PB510B1, System Biosciences) | IDT | | 5'-GATTAT GATCTAGAG TCGCGG CCGCCT G-3' (NotI site for cloning) |
| Antibody | Anti-Phospho-Rb Ser 807/811 (rabbit monoclonal) | Cell Signalling Technology | D20B12 XP, Lot8, RRID:AB_11178658 | 1:500 (FACS) 1:100 (IF) |
| Antibody | Anti-p21 Waf1/Cip1 (rabbit monoclonal) | Cell Signalling Technology | 2947S, Lot 10, RRID:AB_823586 | 1:200 (IF) |
| Antibody | Anti-53BP1 (mouse monoclonal) | BD Biosciences | 612523, Lot 6217571, RRID:AB_399824 | 1:200 (IF) |

*Continued on next page*

*Continued*

| Reagent type (species) or resource | Designation | Source or reference | Identifiers | Additional information |
|---|---|---|---|---|
| Antibody | Anti-RPA2 (mouse monoclonal) | Abcam | Abcam, ab2175, Lot GR3224197-5), RRID:AB_302873 | 1:250 (IF) |
| Antibody | AlexaFluor 488 (donkey anti rabbit) | Abcam | ab150073, Lot GR328726-1, RRID:AB_2636877 | 1:500 for pRb (FACS) |
| Antibody | AlexaFluor 488 (goat anti mouse) | Thermo Fisher Scientific | A11029, Lot 1789729, RRID:AB_138404 | 1:500 for 53BP1 and RPA2 (IF) |
| Antibody | AlexaFluor 647 (goat anti rabbit) | Thermo Fisher Scientific | A21244, Lot 1156625, RRID:AB_141663 | 1:500 for pRb (IF) |
| Antibody | AlexaFluor 568 (donkey anti rabbit) | Thermo Fisher Scientific | A10042, Lot 1134929, RRID:AB_2534017 | 1:500 for p21 in (IF) |
| Commercial assay or kit | Click-IT EdU AlexaFluor 647 HCS Assay | Thermo Fisher Scientific | C10356 | |
| Commercial assay or kit | FxyCycle PI/RNase Staining Solution | Thermo Fisher Scientific | F10797 | |
| Chemical compound, drug | TRIS(2-PYRIDYLMETHYL) AMINE 98% (TPA) | Sigma-Aldrich | 723134 | |
| Chemical compound, drug | N,N,N',N'-Tetrakis (2-pyridylmethyl) ethylenediamine (TPEN) | Sigma-Aldrich | P4413 | |
| Chemical compound, drug | Zinc chloride, anhydrous, 99.95% (metals basis) | Alfa Aesar | 87900 | |
| Chemical compound, drug | Chelex-100, sodium form | Sigma-Aldrch | C7901 | |
| Chemical compound, drug | Neocuproine hydrochloride Monohydrate | Fluka | 72090 | |
| CF4 | Copper-Fluor-4 | Christopher Chang Lab | | PMCID: PMC6008210 |
| Software | BD FACSDiva | BD Biosciences | | Version 8 |
| Software | MATLAB | Mathworks | | V R2017b VR2018a |
| Software | KaleidaGraph | Synergy | | Version 4.02 |
| Software | Prism | GraphPad Software | | Version 8.2.0 |
| Other (Code) | Cell segmentation and tracking pipeline generated in Matlab | | | Desposited: https://biof-git.colorado.edu/biofrontiers-imaging/palmer-z |
| Other | Hoechst 33258 | Sigma-Aldrich | 861405 | Used at 0.1 µg /mL in PBS |

## Cell culture

MCF10A cells were obtained from ATCC and maintained in full growth DMEM/F12 medium (FGM) supplemented with 5% horse serum, 1% Pen/strep antibiotics, 20 ng/mL EGF, 0.5 µg/ml hydrocortisone, 100 ng/ml cholera toxin, and 10 µg/ml insulin in a humidified incubator at 37°C and 5% $CO_2$, as described previously by the Brugge lab (*Debnath et al., 2003*). Cells were passaged with trypsin-EDTA. For imaging and growth experiments, cells were grown in 50:50 Ham's F12 phenol red free/FluoroBrite DMEM with 1.5% Chelex 100-treated horse serum, 1% Pen/strep antibiotics, 20 ng/mL EGF, 0.5 µg/ml hydrocortisone, 100 ng/ml cholera toxin, and 10 µg/ml Chelex 100-treated insulin. Chelex 100 was used to chelate excess $Zn^{2+}$ from horse serum and insulin to generate the defined

minimal medium (MM). ICP-MS was used the measure the elemental content of Chelex 100-treated medium compared to untreated medium and revealed that Chelex 100 treatment significantly reduced Zn and Ni in the medium (*Figure 1—figure supplement 1*). We did not add back Ni (or other elements), but we consistently use Chelex 100–treated MM for all experiments so the concentration of elements other than Zn are the same in all media conditions. ZR medium was generated by supplementing MM with 30 µM $ZnCl_2$ and ZD media was generated by adding 2–3 µM TPA. Cell lines were routinely tested and confirmed to be mycoplasma negative by PCR.

## ICP-MS

ICP-MS was used to measure total Na, Mg, Al, K, Ca, P, S, Mn, Fe, Co, Ni, Cu, and Zn content in defined MM with and without Chelex 100 treatment of horse serum and insulin. Media samples were diluted 1:10 in Chelex 100- treated water with 1% TraceSELECT Ultra nitric acid (free of trace elements). Cells were grown for 24 hr in either 3 µM TPA, 2 µM TPA, or MM and the same elements as above were analyzed (four replicates per condition). Before digestion, cells were counted in triplicate. Cells were pelleted, rinsed with PBS, and cell pellets containing ~6–8 million cells were digested in 100 µL TraceSELECT Ultra nitric acid. Digestion was performed by vortexing constantly for 3 min followed by a 30 min incubation at 70°C. Samples were then diluted to 1 mL in Chelex 100 -treated water. All ICP-MS samples were processed at the Dartmouth Trace Element Analysis Core. Cell ICP-MS results were normalized to average cell counts.

## Plasmids and cell lines

MCF10A cell lines expressing PB-NES ZapCV2 (*Fiedler et al., 2017*) and PB-H2B-mCherry were generated using the PiggyBac Transposon system via electroporation-mediated transfection. Stable cell lines used for long-term imaging were generated with G418 and puromycin antibiotic selection followed by FACS enrichment of dual positive fluorescent cells. The MCF10A cell line stably expressing DHB-Venus (CDK2 sensor) and H2B-mTurquoise2 and the MCF10A p21$^{-/-}$ were provided by the laboratory of Dr. Sabrina Spencer, CU Boulder, and generated as described previously (*Spencer et al., 2013*). Validation of p21 knockout in this cell line was performed using immunofluorescence with a p21 antibody, as described below (See *Figure 6—figure supplement 1*).

## Live cell imaging

Cells were counted with a Countess II Automated Cell Counter (Thermo Fisher Scientific, Waltham, MA) and plated at a density of 2,500–3,500 cells/well ~ 24 hr before imaging in minimal medium in glass bottom 96-well plates (P96-1.5H-N, Cellvis, Mountain View, CA). This starting density was chosen to avoid significant contact inhibition during the imaging period. In *Figures 1*, *2* and *5*, cells were plated in 100 µL MM and 2x media of each specific nutritional regime (ZD, MM, ZR) was added immediately prior to imaging. In *Figure 3*, cells were plated in 100 µL MM and 2x ZD medium was added after 8 hr. In *Figure 7*, cells were plated in 100 µL, 2x medium of specified conditions was added immediately prior to imaging; after 24 hr, 100 µL of medium was removed, and 2x medium of specified conditions was resupplied (effectively half of the medium was replaced with MM or 2x ZR media). For resupply of individual elements in *Figure 7—figure supplement 2*, $ZnCl_2$, $FeCl_2$, or $CuCl_2$ were added at concentrations found in a 50% MM change, 0.73 µM, 0.8 µM, and 0.16 µM, respectively). Images were collected using a Nikon Ti-E inverted microscope microscope with a Lumencor SPECTRA X light engine (Lumencor, Beaverton, OR) and Hamamatsu Orca FLASH-4.0 V2 cMOS camera (Hamamatsu, Japan). Images were collected in time lapse series every 12 mins with a $10 \times 0.45$ NA Plan Apo objective lens (Nikon Instruments, Melville, NY). During imaging, cells were in a controlled environmental chamber surrounding the microscope (Okolab Cage Incubator, Okolab USA INC, San Bruno, CA) at 37°C, 5% $CO_2$ and 90% humidity. Total light exposure time was $\leq 600$ ms per timepoint. Filter sets used for live cell imaging and immunofluorescence (described below) were as follows: CFP Ex: 440 Em: 475/20; GFP Ex: 470, Em: 540/21; YFP Ex: 470, Em: 540/21; CFPYFP FRET Ex: 395, Em: 540/21; mCherry Ex: 555, Em: 595/40; Cy5 Ex: 640, Em: 705/22.

## Image processing/Analysis

A detailed description of our custom MATLAB R2018A (Mathworks) pipeline for automated cell segmentation and tracking, as well as methods for calculating the FRET ratio and CDK2 ratio are

provided in *Supplementary file 1* and the tracking code is available for download here: https://biof-git.colorado.edu/biofrontiers-imaging/palmer-zinc-cell-cycle. Briefly, mitosis events were identified when the nuclear signal generated by fluorescent H2B split into two distinct objects. The FRET ratio (FRET intensity/CFP intensity) was calculated in a cytosolic region outside the nuclear mask and the CDK2 ratio was calculated as the cytosolic intensity/nuclear intensity of the fluorescent CDK2 sensor.

## FACS

Cell pellets were washed with PBS, fixed with 4% paraformaldehyde and permeabilized with methanol at $-20°C$. Cells were stained for one hr with Phospho-Rb Ser 807/811 (D20B12 XP, Cell Signaling Technology, Danvers, MA) at 1:500 dilution prior to Alexa Fluor 488 secondary antibody (ab150073, Abcam, Cambrdige, MA) staining at 1:500 dilution for one hr. Antibodies were diluted in PBST with 1% BSA. PI DNA staining was performed using FxCycle PI/RNAse (Thermo Fisher Scientific F10797). FACS for phospho-Rb and PI was performed on a BD FACSCelesta instrument and analyzed with BD FACSDiva v8 software (BD Biosciences, San Jose, CA). GFP: Ex 488, Em 530/30; mCherry Ex 561, Em 610/20. FACS enrichment of stable cell lines was performed using BD FACSAria Fusion with the following optics: CFP: Ex 445, Em 470/15; YFP: Ex 488, Em 530/30; and mCherry Ex 561, Em 610/20.

## Immunofluorescence

Cells were fixed with 4% paraformaldehyde, washed with PBS, and permeabilized with 0.2% Triton X-100 solution. Blocking was performed in 3% BSA for one hr at 4°C. Primary antibody staining occurred overnight at 4°C, with the following antibody concentrations: p21, 1:200; pRB, 1:100, RPA2, 1:250; 53BP1, 1:200. Following primary staining, cells were washed with PBS and secondary antibody staining was performed for one hr with either AlexaFluor 488, AlexaFluor 568, or Alexa-Fluor 647 each at a 1:500 dilution. Antibodies were diluted in PBS with 3% BSA. Hoechst staining for 15 mins at 0.1 µg/mL diluted in PBS was used to identify nuclei. For experiments comparing CDK2 activity vs. p21 intensity, the fluorescence of the CDK2 sensor was preserved upon fixation. Images were acquired on a Nikon Ti-E inverted microscope (as described in Live Cell Imaging) with a 40 × 0.95 NA Plan Apo Lamda (Nikon). Primary antibodies used were: p21 Waf1/Cip1 (CST, 2947S, Lot 10), pRB Ser807/811 (CST, D20B12 XP, Lot8), RPA2 (Abcam, ab2175, Lot GR3224197-5), 53BP1 (BD Biosciences, 612523, Lot 6217571) and secondaries were: AlexaFluor 488 (Abcam, ab150073, Lot GR328726-1), AlexaFluor 568 (Life Sciences A10042, Lot 1134929), or AlexaFluor 647 (Thermo Fisher Scientific, A21244, Lot 1156625). Analyses were performed with custom MATLAB scripts and run in MATLAB R2018a. Briefly, nuclei were segmented using a combination of adaptive thresholding and the watershed algorithm to segment clumps of cells. The nuclear mask was used to calculate the mean nuclear pRB or p21 intensities for each cell. For cells containing the CDK2 sensor, the nuclear mask was used to draw a ring three pixels wide around the nucleus for computing the CDK2 activity (defined as the ratio of cytoplasmic intensity/nuclear intensity) of the cell. For DNA damage experiments, the centroid of each cell (as defined by the nuclear mask) was used to construct a 140 × 140 pixel square around each cell; this enabled the use of MATLABs 'adaptthresh' function for constructing an accurate foci mask for each cell. The foci mask was further refined by filtering out any objects that were not the appropriate size (area 10–100 $px^2$ for RPA2 and 10–200 $px^2$ for 53BP1) or shape (eccentricity <0.6, where 0 is a perfect circle and one is straight line). Cells were scored as being positive for damage when they had one or more foci present.

## EdU assay

Live cells kept at controlled environmental conditions (37°C, 5% $CO_2$, 90% humidity) were labeled with EdU according to the manufacturer's instructions (Thermo Fisher Scientific, C10356). Briefly, cells were labeled with 10 µM EdU for 15 mins, followed by fixation and permeabilization, as described for immunofluorescence. Cells were labeled for 30 mins via a click reaction with Alexa-Fluor 647 azide using $CuSO_4$ as catalyst. FxCycle PI/RNAse was used to quantify DNA content (2N vs 4N). Images were acquired on a Nikon Ti-E inverted microscope system with a 40 × 0.95 NA Plan Apo Lambda (Nikon). Analysis was performed with custom MATLAB scripts and run in MATLAB R2018a. The analysis workflow was the same as for immunofluorescence, with the exception that high residual background EdU staining necessitated background subtraction before quantifying

intensities. Background subtraction was performed by dividing the image into $11 \times 11$ blocks and then using the lowest 5th percentile intensity value for each block as background. Quantification of DNA content was performed by computing the integrated intensity of each cell.

## FRET sensor calibration and analysis

Sensor calibrations of MCF10A cells stably expressing PB-NES-ZapCV2 were performed using the Nikon Ti-E. Cells were grown for 24 hr in either ZD (3 μM TPA), MM, or ZR media. For collection of $R_{rest}$, cells were imaged for CFP-YFP FRET (200 ms exposure) and CFP (200 ms exposure) every 30 s for several mins. To collect $R_{min}$, 50 μM TPA in MM was added and cells were again imaged for several mins. Cells were then washed three times with phosphate, calcium, and magnesium free HEPES-buffered HBSS, pH 7.4, for removal of TPA. Finally, for collection of $R_{max}$, cells were treated with this HBSS buffer with 119 nM buffered $Zn^{2+}$ solution, 0.001% saponin, and 5 μM pyrithione, as previously described (*Carter et al., 2017*). Average $R_{rest}$ and $R_{min}$ were calculated by averaging across the timepoints collected. The maximum FRET ratio achieved after $Zn^{2+}$ addition was used as $R_{max}$. Images were background corrected by drawing a region of interest in a dark area of the image and subtracting the average fluorescence intensity of the background from the average intensity of each cell. FRET ratios for each cell (n = 8 per condition) were calculated with the following equation: (FRET$_{intensity}$ of cell - FRET$_{intensity}$ of background)/(CFP$_{intensity}$ of cell - CFP$_{intensity}$ of background). Dynamic range (DR) of the sensor in each condition was calculating as $R_{max}/R_{min}$. Fractional saturation (FS) of the sensor in each condition was calculated as follows: $(R_{rest}-R_{min})/(R_{max}-R_{min})$. Finally, $Zn^{2+}$ concentrations were estimated by $[Zn^{2+}]=K_D ((R_{rest} - R_{min})/(R_{max} - R_{rest}))^{1/n}$, where $K_D$ = 5300 pM and n = 0.29 (Hill coefficient).

## Statistical analysis

In *Figure 1A*, FRET ratios between cells grown under ZD, MM, or ZR were compared using One-way ANOVA with post-hoc Tukey HSD, performed using KaleidaGraph v4.02. Alpha was 0.05/confidence level was 0.95. Data were plotted in MATLAB v R2017b. On each box, the central mark is the median and the edges of the box are the 25th and 75th percentiles. The whiskers extend to the most extreme data points, excluding outliers, which are plotted individually with + marks. In *Figure 5A*, differences in DNA damage across conditions were assessed using one-way ANOVA with Sidak's multiple comparisons test (six comparisons, alpha = 0.05). Analyses and graphing were performed using GraphPad Prism 8.2.0. For plots, *p<0.05, **p<0.01, ****p<0.0001. For *Figure 1— figure supplement 1*, two-tailed paired t-tests were used to compare MM to MM+Chelex for each element analyzed. Alpha was 0.05/confidence level was 0.95. For plots *p<0.05 and exact values for significant values are listed. Analysis and graphing were performed using GraphPadPrism 8.2.1. For *Figure 2—figure supplement 1*, one-way ANOVAs were used to determine if ZD3 or ZD2 conditions had elemental levels significantly different than MM levels. Dunnett's multiple comparisons test was used with MM as control sample. Alpha was 0.05/confidence level was 0.95. Analysis and plotting was performed using GraphPad Prism 8.2.1. In plots * indicates when ZD3 or ZD2 was significantly different from MM at p<0.05. Exact p values are listed in a chart with the figure.

## CF4 Cu probe experiment

200,000 MCF10A cells were plated in MM in glass bottom six well plates (P06-1.5H-N Cellvis, Mountain View, CA) and grown for 24 hr. Experimental conditions were as follows: cells left in MM, cells treated with 50 μM neocuproine for 30 min, or cells treated with 3 μM TPA for 2 hr. At the end of the treatment period, 2 μM CF4 Cu probe (reconstituted in MeOH) (*Xiao et al., 2018*) was added and cells were incubated for 20 min at 37°C. Prior to imaging, each dish was rinsed 3X with either MM, MM containing 50 μM Neocuproine, or MM containing 3 μM TPA. Finally, images were collected with a $10 \times 0.45$ NA Plan Apo Lambda (Nikon) using a 514 laser with 525–555 nm emission filter on a Nikon Spinning Disc Confocal microscope. The camera was a 2X Andor Ultra 888 EMCDD (Oxford Instruments, UK). The light source was a HBO Arc lamp. During imaging, cells were in a controlled environmental chamber surrounding the microscope (Okolab Cage Incubator, Okolab USA INC, San Bruno, CA) at 37°C, 5% $CO_2$ and 90% humidity. Image J was used to threshold images to segment cells to generate an ROI for each cell. CF4 intensity for each cell ROI was measured. Background intensity was subtracted from each cell. For each condition, n > 200 cells from one well.

## Acknowledgements

We thank Sabrina Spencer and members of her lab (CU Boulder) for helpful cell-cycle discussions, the CDK2 sensor cell line, and the p21$^{-/-}$ cell line. We thank Dr. Theresa Nahreini (Cell Culture Core Facility) for assisting with flow cytometry and Dr. Joseph Dragavon (BioFrontiers Institute Advanced Light Microscopy Core) for assisting with microscopy. We thank Brian Jackson for ICP-MS analysis at the Dartmouth Trace Element Core Facility, which was established by grants from the National Institute of Health (NIH) and National Institute of Environmental Health Sciences (NIEHS) Superfund Research Program (P42ES007373) and the Norris Cotton Cancer Center at Dartmouth Hitchcock Medical Center. This research was supported by an NIH Director's Pioneer Award DP1-GM114863 (AEP), an Anna and John J Sie Foundation Postdoctoral Fellowship (MNL), and a Molecular Biophysics Training Grant T32 GM-065103 (LJD).

## Additional information

### Funding

| Funder | Grant reference number | Author |
|---|---|---|
| National Institutes of Health | DP1-GM114863 | Amy E Palmer |
| National Institutes of Health | T32 GM-065103 | Leah J Damon |
| Anna and John J. Sie Foundation | Postdoctoral fellowship | Maria N Lo |

The funders had no role in study design, data collection and interpretation, or the decision to submit the work for publication.

### Author contributions

Maria N Lo, Conceptualization, Data curation, Formal analysis, Validation, Investigation, Visualization, Methodology, Writing—original draft, Writing—review and editing; Leah J Damon, Data curation, Formal analysis, Investigation, Visualization, Methodology, Writing—original draft, Writing—review and editing; Jian Wei Tay, Software, Methodology, Writing—original draft, Writing—review and editing; Shang Jia, Resources; Amy E Palmer, Conceptualization, Formal analysis, Supervision, Funding acquisition, Methodology, Writing—original draft, Project administration, Writing—review and editing

### Author ORCIDs

Maria N Lo https://orcid.org/0000-0001-7428-3507
Leah J Damon https://orcid.org/0000-0002-0863-0328
Amy E Palmer https://orcid.org/0000-0002-5794-5983

### Decision letter and Author response

Decision letter https://doi.org/10.7554/eLife.51107.sa1
Author response https://doi.org/10.7554/eLife.51107.sa2

## Additional files

### Supplementary files

- Transparent reporting form
- Supplementary file 1. Detailed protocol for cell segmentation, tracking, and analysis in MatLab.

### Data availability

All data generated or analyzed during this study are included in the manuscript and supporting files. Source data files have been provided.

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
