## [Decision Letter]

**Acceptance summary:**

This manuscript addresses the important and unanswered question of how zinc deficiency blocks cell proliferation. This is an intriguing issue because while we know much about how the cell cycle is controlled in response to macronutrients, the mechanisms controlling the cell cycle in response to micronutrients such as metal ions is much less clear. The authors have improved on previous studies by adopting a growth condition of mild zinc deficiency in which cell death is minimal and by tracking the behavior of individual cells. They also used a FRET probe to establish the effect of their growth conditions on the labile zinc pool, which is the key determinant of whether newly synthesized zinc proteins obtain their metal cofactor. The authors' key findings are that zinc deficiency causes cells to arrest in either a quiescent phase or stalled in S phase, that cells stalled in S phase have increased DNA damage (consistent with the importance of zinc to DNA damage response pathways), and that the quiescence of zinc deficiency occurs thru mechanisms independently of DNA damage checkpoint regulation or response pathways to macronutrient starvation. Thus, this work is an important first step in defining micronutrient-responsive pathways of growth control. Overall the work is well done and clearly presented

**Decision letter after peer review:**

Thank you for submitting your article "Single cell analysis reveals multiple requirements for zinc in the mammalian cell cycle" for consideration by *eLife*. Your article has been reviewed by three peer reviewers, and the evaluation has been overseen by a Reviewing Editor and Philip Cole as the Senior Editor. The following individuals involved in review of your submission have agreed to reveal their identity: David Eide (Reviewer #1); Amanda Bird (Reviewer #3).

The reviewers have discussed the reviews with one another and the Reviewing Editor has drafted this decision to help you prepare a revised submission.

Summary:

The manuscript by Lo et al. addresses the important and unanswered question of how zinc deficiency blocks cell proliferation. While we know much about how the cell cycle is controlled in response to macronutrients, the mechanisms controlling the cell cycle in response to micronutrients such as Zinc is much less clear. The authors have improved on previous studies by adopting a growth condition of mild zinc deficiency in which cell death is minimal and by tracking the behavior of individual cells. Using a FRET probe they establish the effect of growth conditions on the labile zinc pool, which is the key determinant of whether newly synthesized zinc proteins obtain their metal cofactor.

Using markers of cell cycle progression, they show that zinc-deficient cells either become quiescent after mitosis or enter the cell cycle and stall in S-phase, and that zinc supplementation triggers re-entry into the cell cycle. Further, when zinc deficient, quiescent cells do not experience increased DNA damage, whereas cycling cells do. This is consistent with the importance of zinc in DNA damage response pathways and suggests that quiescence due to zinc deficiency occurs independent of DNA damage checkpoint regulation or responses to macronutrient starvation.

The experiments are well executed, clearly presented and point to a model where zinc is important at multiple places in the cell cycle. Given that many important zinc binding proteins have conserved roles in DNA synthesis, these are new and important observations that likely have broad significance. This work is an important first step in defining micronutrient-responsive pathways of growth control.

Essential revisions:

Two common major concerns of the reviewers that need to be addressed in a revision are as follows:

1) The use of TPA as the chelator to establish zinc-deficient growth conditions. While the authors state that TPA is zinc specific, chelators are notoriously nonspecific in the metals they can bind and the same may be the case with TPA. If TPA is binding other metals, the effects observed may be unrelated to zinc deficiency or possibly a combination of zinc deficiency and deficiency for some other metal nutrient. The authors should better document the properties of TPA and its potential for disrupting other metals. The reviewers suggest ICP-MS of cells grown with and without TPA to look at effects on other metals to address TPA specificity.

2) Chelex 100 will bind many other transition metals, including iron and manganese, besides zinc. Given that some of these transitions metals are also required for progression through the cell cycle (e.g. iron is needed for ribonucleotide reductase), it is possible that low levels of other metals may influence the outcome of the performed studies. It was unclear whether other metal ions were added back to the growth medium following Chelex treatment. Does Chelex treatment lead to a significant reduction in the levels of any other metals other than zinc, and if so where these metals added back? If not, do the profiles change if these nutrients are returned back to the growth medium? The reviewers suggest ICP-MS analysis as a way to address this issue as well.

---

## [Author Response]

Essential revisions:Two common major concerns of the reviewers that need to be addressed in a revision are as follows:1) The use of TPA as the chelator to establish zinc-deficient growth conditions. While the authors state that TPA is zinc specific, chelators are notoriously nonspecific in the metals they can bind and the same may be the case with TPA. If TPA is binding other metals, the effects observed may be unrelated to zinc deficiency or possibly a combination of zinc deficiency and deficiency for some other metal nutrient. The authors should better document the properties of TPA and its potential for disrupting other metals. The reviewers suggest ICP-MS of cells grown with and without TPA to look at effects on other metals to address TPA specificity.

The reviewers are indeed correct that most chelators bind a variety of metals, albeit with different affinities. To address this point, we did the following: 1) we surveyed the literature for information on the binding of TPA to other metals and provide a summary in Figure 1—figure supplement 2. 2) We tested whether the low concentrations of TPA used in this manuscript (3 μM) could alter the labile copper pool in MCF10A cells (Figure 1—figure supplement 2). 3) We measured ICP-MS of cells grown in TPA for 24 hrs (Figure 2—figure supplement 1), as suggested by the reviewers. The results of these are discussed below.

a) The only paper we could find measurement of binding of TPA to a variety of metal ions was: von Giorgio Anderegg et al., *Helvetica Chimica Acta*, Vol 60, Fasc 1 (1977) – Nr 1. We converted the ΔG_binding_ values reported in this paper to K_association_ and K_dissociation_ values (Figure 1—figure supplement 2; assuming standard state conditions and 293K). The K_d_ for binding to Zn^2+^ was 4.95 x 10^-12^ M. Given that the labile Zn^2+^ pool in cells is ~ 100 pM (from multiple literature reports), TPA would be expected to decrease the labile Zn^2+^ pool (as demonstrated in Figure 1). The Kd of TPA for other biologically relevant metal ions such as Fe and Mn were 3 to 6 orders of magnitude weaker, respectively, suggesting that TPA would be unlikely to perturb those metals. To our surprise, TPA had a high affinity for Cu^1+^ (K_d_ 2.24 x 10^-17^ M). Multiple studies indicate that there is a kinetically labile pool of Cu in cells (Ackerman and Chang, 2017) and recent estimates suggest the level of Cu^1+^ is 10^-18^ M (Morgan et al., 2019). Multiple studies have indicated that 50-200 μM bathocuproine sulfate (BCS) or neocuproine (K_d_ for Cu^1+^ 1.26 x 10^-20^ M) is required to reduce the labile Cu^1+^ pool (Dodani et al., PNAS, 108(15), 2011; Xiao et al., 2018). Thus, we speculated that 2 – 3 μM TPA (with a 1000-fold weaker affinity than BCS or neocuproine for Cu^1+^) would be unlikely to directly perturb the Cu^1+^ pool. To address this experimentally, we carried out the experiment detailed in point #2 below.

b) To test whether 3 μM TPA can perturb the labile Cu^1+^ pool in cells, we obtained CF4 (K_d_ 2.9 x 10^-13^ M; Xiao et al., 2018) from Professor Chris Chang, treated cells with 3 μM TPA for two hours (conditions known to significantly decrease the labile Zn^2+^ pool), and added CF4 to measure the labile Cu^1+^ pool. As shown in Figure 1—figure supplement 2, 3 μM TPA did not cause a measurable change in the CF4 fluorescence signal, while treatment with the Cu^1+^ chelator neocuproine under typical treatment conditions (50 μM, 30 min) did cause a decrease in signal, demonstrating that CF4 is capable of detecting decreases in the labile Cu^1+^ pool in MCF10A cells. These experiments suggest that although TPA can bind Cu^1+^ with high affinity, the low concentrations of TPA used in this study don’t directly perturb the labile Cu^1+^ pool.

c) As suggested by the reviewers, we carried out ICP-MS on cells treated with 2 or 3 μM TPA for 24 hrs. Importantly, these cells are quiescent and multiple studies have shown that quiescent cells undergo major transcriptional changes. We did observe significant changes in element distribution in these cells (Figure 2—figure supplement 1). We believe that these changes result from remodeling of homeostasis in quiescent cells, perhaps as a consequence of zinc deficiency. For example, zinc deficiency may lead to changes in expression of zinc regulatory proteins such as transporters, and these transporters can also transport other metal ions. We think it is unlikely these changes occur because of direct perturbation of these elements by TPA, particularly given our experiments in (2) and because some elements (e.g. Mn) actually increase. But, to further strengthen the point that the main driver of quiescence is zinc depletion, we tested whether addition of zinc alone could rescue the quiescent phenotype as detailed in point #4 below.

d) In order to demonstrate the deficiency of zinc was the main driver of quiescence induction, we tested whether Zn alone, Cu alone, or Fe alone could rescue the quiescent state and induce cell cycle re-entry. As shown in Figure 7, minimal media was sufficient to induce cell cycle re-entry. Specifically, we replaced half the media with either minimal media or zinc-rich media. To test whether other elements present in minimal media could rescue quiescence, we added the concentration of each element measured by ICP-MS (Zn: 0.73 μM, Cu: 0.16 μM, Fe: 0.8 μM) to test for rescue. As shown in Figure 7—figure supplement 1, only addition of Zn^2+^ was able to rescue the quiescent state. This result reveals that although there are significant changes to cellular ion homeostasis in quiescence, rescue of zinc deficiency is sufficient to induce cell cycle re-entry.

2) Chelex 100 will bind many other transition metals, including iron and manganese, besides zinc. Given that some of these transitions metals are also required for progression through the cell cycle (e.g. iron is needed for ribonucleotide reductase), it is possible that low levels of other metals may influence the outcome of the performed studies. It was unclear whether other metal ions were added back to the growth medium following Chelex treatment. Does Chelex treatment lead to a significant reduction in the levels of any other metals other than zinc, and if so where these metals added back? If not, do the profiles change if these nutrients are returned back to the growth medium? The reviewers suggest ICP-MS analysis as a way to address this issue as well.

We thank the reviewer for raising this point. To address the question, we carried out ICP-MS on Chelex-100 treated and untreated minimal media to define the metal content. The results are presented in Figure 1—figure supplement 1. The data show that the only significant changes in metal content are decreases in zinc (~143 PPB to 95 PPB) and Ni (1 PPB to 0.7 PPB). We also clarify in the Materials and methods that we don’t add back any metals to the Chelex-100-treated minimal media. This means that the Ni concentration is slightly lower than normal. However, we would like to emphasize that we consistently compare TPA-treated or zinc-rich media to the same Chelex-treated minimal media so all medias should have the same concentration of Ni (as well as other nutrients that remained unchanged upon Chelex treatment).